# Generalized Block-Diagonal Structure Pursuit: Learning Soft Latent Task Assignment against Negative Transfer

**Zhiyong Yang**[1,2]    **Qianqian Xu**[3]    **Yangbangyan Jiang**[1,2]
**Xiaochun Cao**[1,2,6]    **Qingming Huang**[3,4,5,6*]

[1]State Key Laboratory of Information Security, Institute of Information Engineering, CAS
[2]School of Cyber Security, University of Chinese Academy of Sciences
[3]Key Lab. of Intelligent Information Processing, Institute of Computing Technology, CAS
[4]School of Computer Science and Tech., University of Chinese Academy of Sciences
[5]Key Laboratory of Big Data Mining and Knowledge Management, CAS
[6]Peng Cheng Laboratory
yangzhiyong@iie.ac.cn, xuqianqian@ict.ac.cn, jiangyangbangyan@iie.ac.cn
caoxiaochun@iie.ac.cn, qmhuang@ucas.ac.cn

## Abstract

In multi-task learning, a major challenge springs from a notorious issue known as *negative transfer*, which refers to the phenomenon that sharing the knowledge with dissimilar and hard tasks often results in a worsened performance. To circumvent this issue, we propose a novel multi-task learning method, which simultaneously learns latent task representations and a block-diagonal Latent Task Assignment Matrix (LTAM). Different from most of the previous work, pursuing the Block-Diagonal structure of LTAM (assigning latent tasks to output tasks) alleviates negative transfer via punishing inter-group knowledge transfer and sharing. This goal is challenging since our notion of Block-Diagonal Property extends the traditional notion for homogeneous and square matrices. In this paper, we propose a spectral regularizer which is proven to leverage the expected structure. Practically, we provide a relaxation scheme which improves the flexibility of the model. With the objective function given, we then propose an alternating optimization method, which reveals an interesting connection between our method and the optimal transport problem. Finally, the method is demonstrated on a simulation dataset, three real-world benchmark datasets and further applied to two personalized attribute learning datasets.

## 1 Introduction

Multi-Task Learning (MTL) is a learning paradigm whose aim is to leverage useful information contained in multiple related tasks to help improve the generalization performance of all the tasks [Caruana, 1997]. Nowadays, MTL has emerged as a fundamental building block for a wide range of applications ranging from scene parsing [Xu et al., 2018], attribute learning [Cao et al., 2018, Yang et al., 2019a, 2018], text classification [Liu et al., 2017], sequence labeling [Lin et al., 2018], to travel time estimation [Li et al., 2018], *etc*.

The fundamental belief of MTL lies in that sharing knowledge among multiple tasks often results in an improvement in generalization performance, which is especially of great significance in the

---

presence of insufficient data annotations [Heskes, 1998]. Based on the belief, a great number of studies have been carried out to explore the problem of how to share valuable knowledge across different tasks. The early studies of MTL (e.g.[Argyriou et al., 2008a]) hold that all the tasks share common and sparse features. However, [Kang et al., 2011] later points out that if not all the tasks are indeed related, then sharing common features with dissimilar and hard tasks often results in performance degradation, which is termed as *negative transfer*.

To address this issue, recent studies in the odyssey against *negative transfer* fall in two major directions. One line of the researches leverages the grouping effect based on the *latent-task-agnostic* idea which develops structural regularizers where only the original per-task parameters are utilized. [Kang et al., 2011, Kshirsagar et al., 2017] directly formulate the tasking grouping as a mixed integer programming (or a relaxation [Frecon et al., 2018]), which simultaneously learns the group index and the model parameters. [Argyriou et al., 2008b, Zhou et al., 2011a, Jacob et al., 2009, Lee et al., 2016, Liu and Pan, 2017, McDonald et al., 2014] leverage the tasking grouping via enforcing a specific structure, hopefully block-diagonal, on the task correlation matrix. As an extension of this formulation [Zhong and Kwok, 2012] resorts to feature-specific task clustering. The other line of researches formulates the MTL based on the latent task, where the model parameter is represented as a linear combination of latent task basis. [Kumar and III, 2012] gives an early trial of this formulation in search of a more flexible MTL model. Similarly, in the work of [Maurer et al., 2013], a sparse coding model is proposed for MTL, where the dictionary is set as the latent task basis and the code is set as the linear combination coefficients of such basis. Recently, [Lee et al., 2018] also provides an asymmetric learning framework based on the latent task representation where transferring knowledge from unreliable tasks to reliable tasks is explicitly punished.

The two aforementioned directions, i.e., learning grouped model structure and latent task representation provide complementary functions in a sense that the former one avoids inter-group negative transfer, while the latter one focuses on learning a more flexible model. However, the related studies on how to bridge the efforts of these two directions are sparse. To the best of our knowledge, the only two studies along this direction are [Crammer and Mansour, 2012, Barzilai and Crammer, 2015]. However, both studies adopt a strong assumption that each group of tasks is only assigned with one latent task basis.

To leverage a flexible grouping structure with latent task representations, we should allow each task cluster to have multiple latent tasks. Motivated by this, we study the structural learning problem of how to learn a block-diagonal Latent Task Assignment Matrix (LATM). With the block-diagonal structure, tasks within each group share a subset (not necessarily one) of the latent task basis. Since LATM is not a squared matrix and marginal constraints are also necessary to avoid isolated tasks/latent tasks, our notion of block-diagonality generalizes the one adopted in the self-expression scenario [Lu et al., 2019, Lee et al., 2016, Liu and Pan, 2017] , which makes traditional structural regularizers not available to solve our problem. Our first contribution then comes as an equivalent spectral condition that realizes our pursuit of the generalized block-diagonal structure. Then we propose a new MTL method named *Generalized Block-Diagonal Structural Pursuit (GBDSP)*, which utilizes the spectral condition as a novel regularizer with a relaxation scheme. In our optimization method, the intermediate solution produced provides new insights into how negative transfer is alleviated in our model. Theoretical studies show how the proposed regularizer guarantees the expected structure. Finally, empirical studies demonstrate the effectiveness of the proposed method.

## 2 Generalized Block-Diagonal Structure Pursuit

**Notations** The notations adopted in this paper are enumerated as follows. $\mathbb{S}_m$ denotes the set of all symmetric matrices in $\mathbb{R}^{m \times m}$. The eigenvalues of a symmetric matrix $A \in \mathbb{S}_m$ are denoted as $\lambda_1(A), \cdots, \lambda_m(A)$ such that $\lambda_1(A) \geq \lambda_2(A) \geq \cdots \geq \lambda_m(A)$. $\langle \cdot, \cdot \rangle$ denotes the inner product for two matrices or two vectors. Given two vectors $a$ and $b$, $a \oplus b$ denotes the outer sum $a\mathbf{1}^\top + \mathbf{1}b^\top$.

Given two matrices $A$ and $B$, $A \oplus B$ denotes the direct sum of two matrices, i.e., $A \oplus B = \begin{bmatrix} A & 0 \\ 0 & B \end{bmatrix}$,

and we say $A \succeq B$, if $A - B$ is positive semi-definite. For distributions, $\mathcal{N}(\mu, \sigma^2)$ denotes the normal distribution. $\mathcal{P}_m$ denotes the set of all permutation matrices in $\mathbb{R}^{m \times m}$. For two matrices $A$ and $B$ having the same size, $d(A, B) = \|A - B\|_F^2$. Given an event $\mathcal{A}$, $\delta(\mathcal{A})$ denotes the corresponding indicator function. *Moreover, let us note two notations in our paper that are prone*

*to be confused. $k$ denotes the dimension of the latent task representation. $K(K \leq k)$ denotes the number of given groups.*

## 2.1   Model Formulation

Before entering our new method, we first provide a brief introduction of the multi-task learning setting we adopted in this paper. Here we adopt the latent task representation framework proposed in [Kumar and III, 2012]. Given $T$ tasks to be learned simultaneously, we denote the training data as: $\left\{ (\boldsymbol{X}^{(1)}, \boldsymbol{Y}^{(1)}), \cdots, (\boldsymbol{X}^{(T)}, \boldsymbol{Y}^{(T)}) \right\}$. Here $\boldsymbol{X}^{(i)} = [\boldsymbol{X}_1^{(i)}, \cdots, \boldsymbol{X}_{n_i}^{(i)}]^\top$, where $\boldsymbol{X}_j^{(i)} \in \mathbb{R}^{d \times 1}$ is the input feature for the $j$-th sample of the $i$-th task, $n_i$ denotes the number of instances and $d$ represents the feature dimension. Similarly $\boldsymbol{Y}^{(i)} = [Y_1^{(i)}, \cdots, Y_{n_i}^{(i)}]^\top \in \mathbb{R}^{n_i \times 1}$, where $Y_j^{(i)}$ is the corresponding response for the $j$-th sample of the $i$-th task. Following the standard multi-task learning paradigm, we learn a model $\hat{Y}^{(i)}(x) = \boldsymbol{W}^{(i)^\top} x$ to estimate the output response for each task $i$. Here we call $\boldsymbol{W} = [\boldsymbol{W}^{(1)}, \cdots, \boldsymbol{W}^{(T)}] \in \mathbb{R}^{d \times T}$ the *per-task parameter matrix*. Furthermore, to model the relationship among the tasks, we assume that the per-task parameters lie in a low dimensional subspace. To this end, we introduce a set of latent task basis $\boldsymbol{L} \in \mathbb{R}^{d \times k}$, where $k < T$. For each given task $i$, its parameter is then represented as a linear combination of the basis by letting $\boldsymbol{W}^{(i)} = \boldsymbol{L} \boldsymbol{S}^{(i)}$, where $\boldsymbol{S}^{(i)} \in \mathbb{R}^{k \times 1}$ is the combination coefficients. Given a loss function $\ell(y, \hat{y})$, the empirical risk for the $i$-th task is defined as $\mathcal{J}^{(i)} = \sum_{j=1}^{n_i} \ell(Y_j^{(i)}, \hat{Y}_j^{(i)})$. Given proper regularizers $\Omega(\boldsymbol{L}), \Omega(\boldsymbol{S})$, [Kumar and III, 2012] learns $\boldsymbol{L}, \boldsymbol{S}$ from the problem $\operatorname{argmin}_{\boldsymbol{L}, \boldsymbol{S}} \sum_{i=1}^T \mathcal{J}^{(i)} + \alpha_1 \cdot \Omega(\boldsymbol{L}) + \alpha_2 \cdot \Omega(\boldsymbol{S})$. In this paper, we adopt the $F$-norm penalty for $\boldsymbol{L}$, i.e., we set $\Omega(\boldsymbol{L}) = \|\boldsymbol{L}\|_F^2$. And we seek new solutions against negative transfer from $\Omega(\boldsymbol{S})$. In this setting, we must deal with both the latent task representations and the true tasks. To differentiate the two, we refer the former ones to latent tasks $(\boldsymbol{l}_1, \cdots, \boldsymbol{l}_k)$ and the latter ones to output tasks $(\boldsymbol{o}_1, \cdots, \boldsymbol{o}_T)$.

With the latent task formulation $\boldsymbol{W} = \boldsymbol{L}\boldsymbol{S}$, $\boldsymbol{S}$ then captures the importance of the latent tasks to the output tasks. In a natural sense, we regard $S_{i,j}$ as $\mathbb{P}(\boldsymbol{l} = i | \boldsymbol{o} = j)$, namely the possibility of choosing $\boldsymbol{l}_i$ to represent $\boldsymbol{o}_j$. In this probabilistic view, $\boldsymbol{L}\boldsymbol{S}^{(i)}$ now becomes $\mathbb{E}_{\boldsymbol{l} | \boldsymbol{o} = i}(\boldsymbol{L})$, i.e., the expectation of the latent tasks representations assigned to task $\boldsymbol{o}_i$. We then call $\boldsymbol{S}$ the Latent Task Assignment Matrix (LTAM), since the conditional possibility could be considered as a soft assignment score. Before developing a proper regularizer, we must first answer the question that *can we choose $\boldsymbol{S}$ arbitrarily*? Unfortunately, we will immediately see that the answer is negative. Let us denote $\boldsymbol{S}^\ddagger \in \mathbb{R}^{k \times T}$ by $S_{i,j}^\ddagger = \mathbb{P}(\boldsymbol{l} = i, \boldsymbol{o} = j)$, the joint distribution of $\boldsymbol{l}$ and $\boldsymbol{o}$. Note that $\boldsymbol{S}^\ddagger \mathbf{1}$ and $\boldsymbol{S}^{\ddagger^\top} \mathbf{1}$ are marginal distributions on $\boldsymbol{l}$ and $\boldsymbol{o}$, we come to two extreme cases that must be ruled out from consideration. If $(\boldsymbol{S}^\ddagger \mathbf{1}_T)_i = 0$ then $\boldsymbol{l}_i$ becomes an isolated latent task which is irrelevant with all the output task. Similarly, if $(\boldsymbol{S}^{\ddagger^\top} \mathbf{1}_k)_j = 0$ then $\boldsymbol{o}_j$ becomes isolated with no latent tasks assigned to it. To remove extreme cases of such kinds, we then pose normalization constraints on $\boldsymbol{S}^\ddagger$ for each row and column in the form: $\boldsymbol{S}^\ddagger \mathbf{1}_T = \mathbf{a} > \mathbf{0}_k$, $\boldsymbol{S}^{\ddagger^\top} \mathbf{1}_k = \mathbf{b} > \mathbf{0}_T$. To maintain fairness, we do not expect to introduce extra bias from the choice of marginal distribution. Such a spirit guides us to put out $\mathbf{a} = \mathbf{1}_k / k$, $\mathbf{b} = \mathbf{1}_T / T$. Moreover, this also simplifies the relation between $\boldsymbol{S}$ and $\boldsymbol{S}^\ddagger$ with $\boldsymbol{S} = T\boldsymbol{S}^\ddagger$. From all above, we adopt the transportation polytopes $\Pi(\mathbf{a}, \mathbf{b}) = \left\{ \boldsymbol{S}^\ddagger \in \mathbb{R}_+^{k \times T} : \boldsymbol{S}^\ddagger \mathbf{1} = \mathbf{a},\ \boldsymbol{S}^{\ddagger^\top} \mathbf{1} = \mathbf{b} \right\}$ as the feasible set for the parameter $\boldsymbol{S}^\ddagger$.

So far we have known that $\boldsymbol{S}$ must satisfy the marginal constraints to make the solution non-trivial. Now let us step further to seek out what else we should pose on $\boldsymbol{S}$ to suppress inter-group transfer. In this paper, we adopt a basic assumption that the latent tasks and output tasks could be clustered into $K$ independent groups. In order to avoid negative transfer, we hope the possibility to assign $\boldsymbol{l}_i$ to $\boldsymbol{o}_j$ is nonzero if and only if $(i, j)$ belongs to the same group. This leads to a block-diagonal structure of $\boldsymbol{S}^\ddagger$ up to row and column permutations. Next, we give a formal definition of the desired block-diagonal structure with $\boldsymbol{S}^\ddagger \in \Pi(\mathbf{a}, \mathbf{b})$, based on the following simple idea. If the columns and rows of $\boldsymbol{S}^\ddagger$ could be partitioned into $K$ groups, $\boldsymbol{S}^\ddagger$ could then be expressed as a direct sum of $K$ blocks up to proper column and row permutations. The maximum of such $K$ then implies the number of groups in the matrix.[1] This motivates the following definition of the grouping structure, which is termed as the *Generalized K Block Diagonal Property* (GKBDP) in our paper.

**Definition 1** (Generalized $K$ Block Diagonal Property). *Given a matrix $\boldsymbol{S}^\ddagger \in \Pi(\mathbf{a}, \mathbf{b})$, if there exists a permutation matrix over rows $\boldsymbol{P}_r \in \mathcal{P}_k$ and a permutation matrix over columns $\boldsymbol{P}_c \in \mathcal{P}_T$ such that: $\boldsymbol{P}_r \boldsymbol{S}^\ddagger \boldsymbol{P}_c = \bigoplus_{i=1}^K \hat{\boldsymbol{S}}^{(i)}$ where $\hat{\boldsymbol{S}}^{(i)} \neq \boldsymbol{0}$, $\hat{\boldsymbol{S}}^{(i)} \in \mathbb{R}^{k_i \times T_i}$, $\sum_i k_i = k$, $\sum_i T_i = T$, then we define $\chi_{\boldsymbol{S}^\ddagger}(\boldsymbol{P}_r, \boldsymbol{P}_c) = K$. Moreover, we say $\boldsymbol{S}^\ddagger$ is Generalized $K$ Block Diagonal if $\chi_{\boldsymbol{S}^\ddagger} = \max_{\boldsymbol{P}_r \in \mathcal{P}_d, \boldsymbol{P}_c \in \mathcal{P}_T} \chi_{\boldsymbol{S}^\ddagger}(\boldsymbol{P}_r, \boldsymbol{P}_c) = K$.*

Note that GKBDP extends the notion of block-diagonal property deployed in self-expression [Lee et al., 2016, Liu and Pan, 2017, Lu et al., 2019, Jiang et al., 2018, 2019, Yang et al., 2019b] which is only available for square matrices. Furthermore, the traditional self-expression-based block-diagonal property requires $\boldsymbol{P}_r = \boldsymbol{P}_c^\top$ [Lu et al., 2019], i.e., a simultaneous permutation on columns and rows (the $i$-th row and the $i$-th column represent the same object) . However, this is not the case in our notion of GKBDP since the columns and rows here represent heterogeneous concepts (the $i$-th row is for $\boldsymbol{l}_i$, $i$-th column is for $\boldsymbol{o}_i$). Moreover we also consider the marginal constraints $\boldsymbol{S}^\ddagger \mathbf{1} = \mathbf{a}$, and $\boldsymbol{S}^{\ddagger^\top} \mathbf{1} = \mathbf{b}$ to avoid isolated $\boldsymbol{l}$ or $\boldsymbol{o}$. Based on the aforementioned facts, traditional regularization schemes are thus not directly applicable to leverage the GKBDP.

Next, we derive an equivalent condition for GKBDP, which directly leads to the formulation of our method. First, we define an auxiliary bipartite graph $\mathcal{G}_{l \cup o} = (\mathcal{V}_{l \cup o}, \mathcal{E}_{l \cup o}, \boldsymbol{A}_{l \cup o})$. The vertices of $\mathcal{G}_{l \cup o}$ include all $\boldsymbol{l}$ and $\boldsymbol{o}$. Denote $\mathcal{V}_o$ as the set of all output tasks and $\mathcal{V}_l$ as the set of all latent tasks, the vertex set $\mathcal{V}_{l \cup o}$ is then defined as $\mathcal{V}_{l \cup o} = \mathcal{V}_l \cup \mathcal{V}_o$. To define the edge set, we first define an affinity matrix $\boldsymbol{A}_{l \cup o}$ in the form $\boldsymbol{A}_{l \cup o} = \begin{bmatrix} \boldsymbol{0} & \boldsymbol{S}^\ddagger \\ \boldsymbol{S}^{\ddagger\top} & \boldsymbol{0} \end{bmatrix}$, where the $(i, j) \in \mathcal{E}_{l \cup o}$ if and only if $\boldsymbol{A}_{l \cup o ij} \neq 0$. Then the well-known graph Laplacian follows as $\Delta(\boldsymbol{S}^\ddagger) = diag(\boldsymbol{A}_{l \cup o} \mathbf{1}) - \boldsymbol{A}_{l \cup o}$. With the definition of $\mathcal{G}_{l \cup o}$, we could derive the following theorem.

**Theorem 1.** *If $\boldsymbol{S}^\ddagger \in \Pi(\mathbf{a}, \mathbf{b})$, $\chi_{\boldsymbol{S}} = K$ holds if and only if $\dim(Null(\Delta(\boldsymbol{S}^\ddagger))) = K$, i.e, the 0 eigenvalue of $\Delta(\boldsymbol{S}^\ddagger)$ has multiplicity $K$. Moreover, denote $\mathcal{A}^{(i)}$ as the set of latent and output tasks belonging to the $i$-th block of $\boldsymbol{S}$, the eigenspace of 0 is spanned by $\iota_{\mathcal{A}^{(1)}}, \iota_{\mathcal{A}^{(2)}}, \cdots, \iota_{\mathcal{A}^{(K)}}$, where $\iota_{\mathcal{A}^{(i)}} \in \mathbb{R}^{(k+T) \times 1}$, $[\iota_{\mathcal{A}^{(i)}}]_j = 1$ if $j \in \mathcal{A}^{(i)}$, otherwise $[\iota_{\mathcal{A}^{(i)}}]_j = 0$.*

The proof can be found in Appendix B.1. With the theorem, we can now step further to seek a suitable regularizer realizing GKBDP. It becomes straightforward that leveraging GKBDP requires the sum of bottom $K$ eigenvalues to be as small as possible. Let $N = k + T$ denote the total number of nodes in $\mathcal{G}_{l \cup o}$, we then need to minimize $\sum_{N-K+1}^N \lambda_i(\Delta(\boldsymbol{S}^\ddagger))$ with the constraint $\boldsymbol{S}^\ddagger \in \Pi(\mathbf{a}, \mathbf{b})$. Following the variational characterization of eigenvalues, we could reformulate eigenvalue calculation as an optimization problem with the following theorem.

**Theorem 2.** *Let $\mathcal{M} = \{\boldsymbol{U} : \boldsymbol{U} \in \mathbb{S}_N, \boldsymbol{I} \succeq \boldsymbol{U} \succeq \boldsymbol{0}, tr(\boldsymbol{U}) = K\}$, then $\forall \boldsymbol{A} \in \mathbb{S}^N$: $\sum_{N-K+1}^N \lambda_i(\boldsymbol{A}) = \min_{\boldsymbol{U} \in \mathcal{M}} \langle \boldsymbol{A}, \boldsymbol{U} \rangle$, with an optimal value reached at $\boldsymbol{U} = \boldsymbol{V}_K \boldsymbol{V}_K^\top$, where $\boldsymbol{V}_K$ represents the eigenvectors of the smallest $K$ eigenvalues of $\boldsymbol{A}$.*

The theorem slightly extends the results in [Overton and Womersley, 1992b], which considers top-K eigenvalues. A proof for the theorem could be found in Appendix B.2. Back to our practical problem, Thm.2 provides a regularizer as $\Omega(\boldsymbol{S}^\ddagger) = \inf \left\{ \langle \Delta(\boldsymbol{S}^\ddagger), \boldsymbol{U} \rangle : \boldsymbol{U} \in \mathcal{M} \right\}$. Denote $\widetilde{\mathcal{J}} = \sum_{i=1}^T \mathcal{J}^{(i)}$, $\Omega_1 = \alpha_1 \cdot \|\boldsymbol{L}\|_F^2 / 2$, $\Omega_2 = \alpha_3 \cdot \langle \Delta(\boldsymbol{S}^\ddagger), \boldsymbol{U} \rangle$, we then reach an MTL model based on the latent task framework:

$$\min_{\boldsymbol{L}, \boldsymbol{S}, \boldsymbol{S}^\ddagger, \boldsymbol{U}} \widetilde{\mathcal{J}} + \Omega_1 + \Omega_2, \ \ s.t. \ \boldsymbol{S}^\ddagger \in \Pi(\mathbf{a}, \mathbf{b}), \ \boldsymbol{U} \in \mathcal{M}, \ \boldsymbol{S} = T\boldsymbol{S}^\ddagger. \quad (Obj_0)$$

This model exactly realizes GKBDP. However, this exact model is impractical in the following sense. First, it is hard to solve ($Obj_0$) directly since multiple constraints are wrapped together on $\boldsymbol{S}$. Moreover, it encourages a strict structural control of $\boldsymbol{S}$, prohibiting overlapped subspaces even when it benefits the performance. These problems lead us to a relaxed implementation of ($Obj_0$) , bringing us possibilities to embrace a practical and a more flexible solution.

To avoid directly controlling the structure of $\boldsymbol{S}$, we relax the constraint $\boldsymbol{S} = T\boldsymbol{S}^\ddagger$ as a distance penalty[2] $\Omega_3 = \alpha_2 \cdot d(\boldsymbol{S}, T\boldsymbol{S}^\ddagger)/2$. This brings us to the final optimization problem:

$$\min_{\boldsymbol{L}, \boldsymbol{S}, \boldsymbol{S}^{\ddagger}, \boldsymbol{U}} \quad \mathcal{J} = \widetilde{\mathcal{J}} + \Omega_1 + \Omega_2 + \Omega_3, \quad s.t. \ \boldsymbol{S}^{\ddagger} \in \Pi(\mathbf{a}, \mathbf{b}), \ \boldsymbol{U} \in \mathcal{M}. \qquad (Obj)$$

The relaxation scheme improves the flexibility of our model via leveraging a partial structural control, which decomposes $\boldsymbol{S}$ into a structural component $T\boldsymbol{S}^{\ddagger}$ and a dense component as the residual $\boldsymbol{S} - T\boldsymbol{S}^{\ddagger}$. The new dense component allows $\boldsymbol{S}$ to slightly (controlled by the magnitude of $\alpha_2$) violate the structural constraint, in search of a better performance.

Alternatively, we could also interpret $(Obj)$ as a way to leverage hierarchical priors on the model. This is specified by the generative model in the following:

$$\left[\boldsymbol{Y}^{(i)} \mid \boldsymbol{X}^{(i)}, \boldsymbol{L}, \boldsymbol{S}^{(i)}\right] \sim \mathcal{N}\left(\boldsymbol{X}^{(i)}\boldsymbol{L}\boldsymbol{S}^{(i)}, \boldsymbol{I}\right), \qquad \left[vec(\boldsymbol{L}) \mid \alpha_1\right] \sim \mathcal{N}\left(\boldsymbol{0}, \frac{1}{\alpha_1}\boldsymbol{I}\right),$$

$$\left[\boldsymbol{S}^{(i)} \mid T\boldsymbol{S}^{\ddagger(i)}, \alpha_2\right] \sim \mathcal{N}\left(T\boldsymbol{S}^{\ddagger(i)}, \frac{1}{\alpha_2}\boldsymbol{I}\right), \qquad \left[\boldsymbol{S}^{\ddagger} \mid \boldsymbol{U}, \alpha_3\right] \sim g, \quad \boldsymbol{U} \sim h,$$

where

$$g \propto \exp\left(-\alpha_3 \cdot \left\langle \Delta(\boldsymbol{S}^{\ddagger}), \boldsymbol{U}\right\rangle\right) \cdot \delta\left(\boldsymbol{S}^{\ddagger} \in \Pi(\mathbf{a}, \mathbf{b})\right), \ h \propto \delta\left(\boldsymbol{U} \in \mathcal{M}\right).$$

Here $g$ specifies an exponential distribution restricted on the set $\Pi(\mathbf{a}, \mathbf{b})$, and $h$ specifies a uniform distribution on the set $\mathcal{M}$. With this process, our objective function is equivalent to a Maximum A Posterior (MAP) formulation in the following sense:

$$\mathcal{J} = -\log\left(\mathbb{P}(\boldsymbol{L}, \ \boldsymbol{S}, \ \boldsymbol{S}^{\ddagger}, \ \boldsymbol{U} \mid \boldsymbol{X}, \ \boldsymbol{Y}, \alpha_1, \alpha_2, \alpha_3)\right) + const.$$

This fact gives us an alternative perspective on the relationship between $\boldsymbol{S}$ and $\boldsymbol{S}^{\ddagger}$. With the relaxation scheme, the constraints are moved to the mean of the prior distribution of $\boldsymbol{S}$. This provides $\boldsymbol{S}$ with a possibility to activate the overlapping off-diagonal block elements with a moderate variance $\alpha_2$.

## 2.2 Optimization

It is easy to see that $\mathcal{J}$ in $(Obj)$ is not a jointly convex function. But fortunately, it is easy to show that the four subproblems with respective to $\boldsymbol{L}, \boldsymbol{S}, \boldsymbol{S}^{\ddagger}, \boldsymbol{U}$ are all convex. Instead of directly solving the overall non-convex problem, this fact motivates us to adopt an alternating optimization scheme where only one of the four parameters is updated each time and the others are fixed as constants. Now we elaborate the four subproblems, respectively.

$\boldsymbol{L}$ and $\boldsymbol{S}$ subroutine: Theoretically, both subroutines solve a strongly convex unconstrained quadratic programming and enjoy a closed-form solution. However, calculating the closed form of the $\boldsymbol{L}$ subproblem suffers from a heavy computational complexity. Instead of adopting the closed form directly for the $\boldsymbol{L}$ subproblem, we adopt a gradient-based optimizer in our paper. Please see Appendix C for more details.

$\boldsymbol{U}$ subroutine: According to Thm.2, $\boldsymbol{U}$ could be solved from: $\boldsymbol{U} = \boldsymbol{V}_K \boldsymbol{V}_K^{\top}$, where $\boldsymbol{V}_K$ denotes eigenvectors associated with the smallest $K$ eigenvalues of $\Delta(\boldsymbol{S}^{\ddagger})$. Denote $\boldsymbol{V}_K = [\boldsymbol{f}_1, \cdots, \boldsymbol{f}_{k+T}]^{\top}$, according to Thm.1, when $\chi_{\boldsymbol{S}^{\ddagger}} = K$, up to some orthogonal transformation, $\boldsymbol{f}_i \in \mathbb{R}^{K \times 1}$ becomes an indicator vector with only one non-zero entry where $[\boldsymbol{f}_i]_j = 1$ only if the corresponding latent/output task $i$ is in group $\iota_{\mathcal{A}(j)}$. In this way, we see that $\boldsymbol{f}_i$ is a strong group indicator. Consequently, we name $\boldsymbol{f}_i$ as the *embedding vector* for the latent (output) task.

$\boldsymbol{S}^{\ddagger}$ subroutine: With $\boldsymbol{U}$ updated with $\boldsymbol{U} = \boldsymbol{V}_K \boldsymbol{V}_K^{\top}$, the following proposition shows a way to solve this subproblem:

**Proposition 1.** *The $\boldsymbol{S}^{\ddagger}$ subproblem could be reformulated as:*

$$\min_{\boldsymbol{S}^{\ddagger} \in \Pi(\mathbf{a}, \mathbf{b})} \ \frac{\vartheta}{2}\|\boldsymbol{S}^{\ddagger} - \bar{\boldsymbol{S}}\|_F^2 + \left\langle \mathcal{D}, \boldsymbol{S}^{\ddagger}\right\rangle, \qquad (Primal)$$

*where $\vartheta = \dfrac{\alpha_2 \cdot T^2}{\alpha_3}$, $\bar{\boldsymbol{S}} = \dfrac{\boldsymbol{S}}{T}$ and $\mathcal{D}_{ij} = \|\boldsymbol{f}_i - \boldsymbol{f}_{k+j}\|^2$.*

The proof could be found in Appendix A.1. From Prop.1, we see that the subproblem recovers a smoothed Optimal Transport (OT) [Peyré et al., 2019] problem. More specifically, the calculation of the Wasserstein-2 distance between $l$ and $o$, based on the spectral embedding.

Similar to the recent results [Blondel et al., 2018, Peyré et al., 2019], we can show that the regularized OT problem has a close connection with the original OT problem.

**Proposition 2.** *The following properties hold true:*

*(a) Denote $\boldsymbol{S}^{\ddagger}{}_{\vartheta_0}$ as the solution of problem ($Primal$) when $\vartheta = \vartheta_0$. Then we have :*

$$\boldsymbol{S}^{\ddagger}{}_{\vartheta_0} \overset{\vartheta_0 \to 0}{\to} \underset{\boldsymbol{S}^{\ddagger}}{\operatorname{argmin}} \left\{ d(\boldsymbol{S}^{\ddagger}, \bar{\boldsymbol{S}}) : \boldsymbol{S}^{\ddagger} \in \underset{\boldsymbol{S}^{\ddagger} \in \Pi(\mathbf{a}, \mathbf{b})}{\operatorname{argmin}} \left\langle D, \boldsymbol{S}^{\ddagger} \right\rangle \right\}.$$

*(b) Denote*

$$\mathcal{J}_{OT} = \min_{\boldsymbol{S}^{\ddagger} \in \Pi(\mathbf{a}, \mathbf{b})} \left\langle \mathcal{D}, \boldsymbol{S}^{\ddagger} \right\rangle, \quad \mathcal{J}_{REG} = \min_{\boldsymbol{S}^{\ddagger} \in \Pi(\mathbf{a}, \mathbf{b})} (\vartheta/2) \cdot d(\boldsymbol{S}, \bar{\boldsymbol{S}}) + \left\langle \mathcal{D}, \boldsymbol{S}^{\ddagger} \right\rangle,$$

*we have:*

$$\vartheta \cdot \max \left\{ d(\bar{\boldsymbol{S}}\mathbf{1} - \mathbf{a})/T, d(\bar{\boldsymbol{S}}^{\top}\mathbf{1}, \mathbf{b})/k \right\} \le \mathcal{J}_{REG} - \mathcal{J}_{OT} \le \vartheta \cdot (\min\{\|\mathbf{a}\|_2^2, \|\mathbf{b}\|_2^2\} + \|\bar{\boldsymbol{S}}\|_F^2).$$

The proof can be found in Appendix A.2. Prop.2-(a) shows that asymptotically, when $\vartheta \to 0$, the solution of the regularized OT problem approaches a specific solution of the original OT problem. More specifically, it will pick out an optimal coupling from the OT solution set with the smallest regularization term $d(\boldsymbol{S}^{\ddagger}, \bar{\boldsymbol{S}})$. From a non-asymptotic perspective, Prop.2-(b) shows how fast this approximation will take place. Consequently, we will get a reasonable approximation of the original OT problem with a small $\vartheta$. Moreover, if the regularizer $\left\langle \Delta(\boldsymbol{S}^{\ddagger}), \boldsymbol{U} \right\rangle$ is sufficiently small, $\boldsymbol{f}$ approaches the grouping indicator of a K-connected bipartite graph. At the same time, $\mathcal{D}_{ij}$, the distance between the embedding vectors, approaches zero when $\boldsymbol{l}_i$ and $\boldsymbol{o}_j$ belong to the same group indicated by $\boldsymbol{f}$. Under this circumstance, the transportation cost $\mathcal{D}_{i,j}$ is small only if $\boldsymbol{l}_i$ and $\boldsymbol{o}_j$ belong to the same group. By contrast, the inter-group negative transfer is suppressed with a large transportation cost. Moreover, with $\alpha_2 \to +\infty$, $\boldsymbol{L}\boldsymbol{S}^{(i)} \to \mathbb{E}_{\boldsymbol{l}|\boldsymbol{o}=i}(\boldsymbol{L})$. This indicates that the conditional expectation also embraces the idea of barycenter projection mapping [Seguy et al., 2018] in the sense $\mathbb{E}_{\boldsymbol{l}|\boldsymbol{o}=i}(\boldsymbol{L}) = \operatorname{argmin}_{\boldsymbol{z}} \mathbb{E}_{\boldsymbol{l}|\boldsymbol{o}=i}(d(\boldsymbol{L}^{(i)}, \boldsymbol{z}))$. Under this condition, the task parameter of $\boldsymbol{o}_i$ becomes a barycenter of the latent task embeddings. Finally, we show that this subproblem could be solved efficiently from the dual formulation.

**Proposition 3.** *The dual problem of ($Primal$) could be solved from:*

$$(\boldsymbol{h}^{\star}, \ \boldsymbol{g}^{\star}) = \underset{\boldsymbol{h}, \ \boldsymbol{g}}{\operatorname{argmin}} \ \frac{1}{2\vartheta} \cdot \left\| (\boldsymbol{h} \oplus \boldsymbol{g} - \mathcal{D} + \vartheta\bar{\boldsymbol{S}})_+ \right\|_F^2 - \left\langle \boldsymbol{h}, \mathbf{a} \right\rangle - \left\langle \boldsymbol{g}, \mathbf{b} \right\rangle, \qquad (Dual)$$

*and the primal solution is given by $\boldsymbol{S}^{\ddagger\star} = \left[ \dfrac{\boldsymbol{h}^{\star} \oplus \boldsymbol{g}^{\star} - \mathcal{D}}{\vartheta} + \bar{\boldsymbol{S}} \right]_+$.*

The proof can be found in Appendix A.3. From Prop.3, we can recover the primal solution from ($Dual$), which only involves $\mathsf{O}(k + T)$ parameters instead of $\mathsf{O}(kT)$. In this spirit, we first solve $\boldsymbol{h}^{\star}, \boldsymbol{g}^{\star}$ from ($Dual$) with the L-BFGS [Zhu et al., 1997] method as the optimizer, and then recover $\boldsymbol{S}^{\ddagger\star}$ from the dual parameters.

**Summary** Our optimization procedure then alternatively solves the four subproblems until a convergence condition is reached, with irrelevant variables fixed as their latest version. Moreover, since all the subproblems are convex, it is easy to see that the iteration over subproblems then keeps the overall loss function non-increasing.

## 2.3 Theoretical Analysis

In this section, we present theoretical analysis shedding light on how the hyperparameters $\alpha_1, \alpha_2, \alpha_3$ affect our proposed model. Let us start with defining a proper hypothesis space $\mathcal{H}$ that covers the solution returned by the optimization algorithm. Recall that ($Obj$) is non-increasing during the

optimization procedure. This means that if we choose a feasible candidate $L_0, S_0, S^{\ddagger}{}_0, U_0$ as the initialization of the algorithm, denote by $\mathcal{J}_0$ the corresponding objective function value, we will have $\|L\|_F^2 \leq 2\mathcal{J}_0/\alpha_1, d(S, TS^{\ddagger}) \leq 2\mathcal{J}_0/\alpha_2, \langle\Delta(S^{\ddagger}), U\rangle \leq 2\mathcal{J}_0/\alpha_3$, for all outputs from the optimization algorithm. This naturally defines a hypothesis class $\mathcal{H} = \mathcal{H}(L, S, S^{\ddagger}, U)$:

$$\mathcal{H}(L, S, S^{\ddagger}, U) = \left\{ \left\{ \hat{Y}^{(i)}(X_i^{(t)}) = (LS^{(i)})^\top X_i^{(t)} \right\}_{ti} : \|L\|_F^2 \leq \xi_1, \right.$$
$$\left. d(S, TS^{\ddagger}) \leq \xi_2, \ \langle\Delta(S^{\ddagger}), U\rangle \leq \xi_3, S^{\ddagger} \in \Pi(\mathbf{a}, \mathbf{b}), U \in \mathcal{M} \right\},$$

where $\xi_1 = 2\mathcal{J}_0/\alpha_1, \xi_2 = 2\mathcal{J}_0/\alpha_2, \xi_3 = 2\mathcal{J}_0/\alpha_3$. Now we are ready to represent the theoretical results based on $\mathcal{H}$.

As the first step, we explore how well a model learned from $\mathcal{H}$ generalizes to the overall population. The empirical risk $\hat{\mathcal{R}}(L, S)$ over the observed dataset and the task-averaged risk $\mathcal{R}(L, S)$ are given as: $\hat{\mathcal{R}}(L, S) = \sum_{i=1}^T \mathcal{J}^{(i)}/T, \mathcal{R}(L, S) = \sum_{i=1}^T \mathbb{E}_{\mu_i}\left[\mathcal{J}^{(i)}\right]/T$, where the training data $X_j^{(i)}, Y_j^{(i)}$ are sampled from $\mu_i$. We then bound the term $\Delta = \mathcal{R}(L, S) - \hat{\mathcal{R}}(L, S)$. Following the spirit of [Maurer et al., 2016], we have the following bound for the hypothesis space:

**Theorem 3.** *Suppose that $n_1 = n_2 \cdots = n_T = n$, the loss function $\ell(y, \cdot) : \hat{y} \mapsto [0, M], \ell(y, \cdot)$ is $M\phi$-Lipschitz continuous, and $\forall (L, S)$ are chosen from $\mathcal{H}$, the following bound holds with possibility at least $1 - \delta$ :*

$$\frac{\Delta}{M} \leq \kappa_1 \phi \aleph \left( \frac{\xi_1 k \|COV(X)\|_1}{nT} \right)^{1/2} + 2\kappa_2 \phi \aleph \cdot \left( \frac{\xi_1 \|COV(X)\|_\infty}{n} \right)^{1/2} + \left( \frac{9\ln(2/\delta)}{2nT} \right)^{1/2},$$

*where $\kappa_1$ and $\kappa_2$ are two universal constants, $\aleph = \sqrt{\xi_2} + 1$. $COV(X)$ is the covariance operator defined as $\langle COV(X)u, v\rangle = (1/nT) \cdot \sum_{ti} \left\langle u, X_i^{(t)} \right\rangle \left\langle X_i^{(t)}, v \right\rangle$.*

The proof can be found in Appendix B.3. With the sample complexity given, we narrow our focus to the problem that how $\xi_2, \xi_3$ benefit the hypothesis space. The following theorem shows that $\xi_2, \xi_3$ control the spectral properties of $S$ and $S^{\ddagger}$.

**Theorem 4** (Spectral Properties of $S$). *Let $k \leq T$, define the SVD of $S$ as $S = P\Lambda Q^\top$, where $P = [p_1, \cdots, p_k], Q = [q_1, \cdots, q_k]$ are left and right singular vectors respectively, $\Lambda = diag(\sigma_i(S))$ with $\sigma_1(S) \geq \sigma_2(S) \cdots \geq \sigma_k(S) \geq 0$. The following properties hold for all $S \in \mathcal{H}$:*

(a) *The bottom $K$ eigenvalues of the graph Laplacian induced by $S$ is bounded by : $\sum_{i=N-K+1}^N \lambda_i(\Delta(S)) \leq T\xi_3 + \sqrt{\xi_2 K}(\sqrt{2} + \sqrt{k} + \sqrt{T})$, where $\Delta(S)$ is obtained from replacing $S^{\ddagger}$ in $\Delta(S^{\ddagger})$ with $S$ .*

(b) *Define $\mathbb{M}(P) = Span\{p_1, \cdots p_K\}$ and $\mathbb{M}(Q) = Span\{q_1, \cdots q_K\}$, if $\xi_3 + \sqrt{\xi_2}/T < 1/T$ and $rank(S) \geq K$, then we have:*

$$\frac{\sigma_1(S)}{\sigma_K(S)} = \max_{\substack{x, y \in \mathbb{M}(P) \\ \|x\|_2 = 1, \|y\|_2 = 1}} \frac{\|Sx\|_2}{\|Sy\|_2} = \max_{\substack{x, y \in \mathbb{M}(Q) \\ \|x\|_2 = 1, \|y\|_2 = 1}} \frac{\|S^\top x\|_2}{\|S^\top y\|_2} \leq \frac{1}{k} \cdot \frac{T + k\sqrt{\xi_2}}{1 - T\xi_3 - \sqrt{\xi_2}}.$$

The proof can be found in Appendix B.4. Thm.4.(a) implies that, with a small $\xi_2$ and $\xi_3$, the grouping structure of $S$ could also be controlled, even though the structural penalty is not directly exhibited on $S$. More specifically, if we pick $\xi_3 = O(T^{-3/2})$ and $\xi_2 = O(1/T^2)$, we can reach a small $\sum_{i=N-K+1}^N \lambda_i(\Delta(S))$ with $O(T^{-1/2})$ if $T >> k$. Thm.4.(b) states that shrinking $\xi_2, \xi_3$ helps to remain a smaller numerical perturbation of $Sx$ ($S^\top x$) over the principle subspaces, i.e., the subspaces spanned by principle left/right singular vectors. Besides numerical benefits, the following theorem shows that $\xi_3$ guarantees good structure recovery in a non-asymptotic manner.

**Theorem 5.** *Assume that $k \leq T$, and that the ground-truth grouping is indicated by $\mathcal{G} = \{(i, j) : 1 \leq i \leq k, 1 \leq j \leq T, l_i \text{ and } o_i \text{ are in the same group}\}$ with $K$ disjoint groups. Moreover, for a matrix $W$, denote $supp(W)$ as $\{(i, j) : W_{i,j} \neq 0\}$. For all $S^{\ddagger}$ obtained from the space $\mathcal{H}$ such*

*that $\lambda_{K+1}(\Delta(\boldsymbol{S}^{\ddagger})) > \lambda_K(\Delta(\boldsymbol{S}^{\ddagger})) > 0$ and $\inf_{\boldsymbol{S}^{\star} \in \Pi(\mathbf{a},\mathbf{b}), Supp(\boldsymbol{S}^{\star})=\mathcal{G}} ||\Delta(\boldsymbol{S}^{\ddagger}) - \Delta(\boldsymbol{S}^{\star})||_F \leq \epsilon$, we have:*

$$\|\boldsymbol{S}^{\ddagger supp^c}\|_1 \leq \frac{1}{2} \cdot \left( \xi_3 + \frac{\sqrt{\frac{2}{k} + \frac{6}{T}} \cdot \epsilon}{\lambda_{K+1}(\Delta(\boldsymbol{S}^{\ddagger}))} \right) \leq \frac{1}{2} \cdot \xi_3 + \frac{4}{\sqrt{kT}\lambda_{K+1}(\Delta(\boldsymbol{S}^{\ddagger}))}$$

$\boldsymbol{S}^{\ddagger supp^c}$ *denotes the projection of $\boldsymbol{S}^{\ddagger}$ onto the complement of the support set of the expected block-diagonal structure in the sense that $\boldsymbol{S}^{\ddagger supp^c}_{i,j} = 0$ if $i$ and $j$ belong to the same group, $\boldsymbol{S}^{\ddagger supp^c}_{i,j} = \boldsymbol{S}^{\ddagger}_{i,j}$ otherwise.*

The proof can be found in Appendix B.5. Under the assumptions of Thm.5, a smaller $\xi_3$ embraces a better recovery of the true block-diagonal structure. More specifically, it shrinks $\|\boldsymbol{S}^{\ddagger supp^c}\|_1$, i.e., the overall magnitude of the elements that violate the true grouping structure. Picking $\xi_3 = \mathsf{O}(T^{-1/4})$ and $k = \mathsf{O}(\sqrt{T})$, if $\lambda_{K+1}(\Delta(\boldsymbol{S}^{\ddagger})) = \mathsf{O}(1/k)$, under the worst case, we have $\|\boldsymbol{S}^{\ddagger supp^c}\|_1 = \mathsf{O}(T^{-1/4})$.

## 3 Empirical Study

### 3.1 Experiment Settings

For all the experiments, hyper-parameters are tuned based on the training and validation set, and the results on the test set are recorded. The experiments are done with 5 repetitions for each involved algorithm. Except for the Simulated Dataset, the train/valid/test ratio is fixed as 70%/15%/15%. For regression datasets (Simulated dataset and School), we adopt the overall rmse on all samples as the evaluation metric. For classification datasets, we adopt the average of task-wise AUC as the evaluation metric. For regression problem, $\mathcal{J}(\cdot)$ in GBDSP is chosen as the square-loss. For classification problem, $\mathcal{J}(\cdot)$ in GBDSP is chosen as the squared surrogate loss for AUC [Gao et al., 2016]. All the experiments are run with MATLAB 2016b and a Ubuntu 16.04 system. In the next subsection, we show our experimental results on a simulated dataset. More experiments for real-world datasets could be found in Appendix D.

### 3.2 Simulated Dataset

To test the effectiveness of GBDSP we generate a simple simulated annotation dataset with $T = 150$ simulated tasks, where the dataset is produced according to the assumption in our model. For each task, 500 samples are generated with $d = 300$ features such that $\boldsymbol{X}^{(i)} \in \mathbb{R}^{500 \times 300}$ and $\boldsymbol{x}_k^{(i)} \sim \mathcal{N}(0, \boldsymbol{I}_{80})$. Specifically, we generate latent task representations with $k = 100$ basis. This yields an $\boldsymbol{L} \in \mathbb{R}^{300 \times 100}$ and an $\boldsymbol{S} \in \mathbb{R}^{100 \times 150}$. To leverage the group structure, we split the latent tasks and output tasks into 5 groups, in a way that $\boldsymbol{L} = [\boldsymbol{L}_1, \cdots, \boldsymbol{L}_5]$, where $\boldsymbol{L}_1 \in \mathbb{R}^{300 \times 20}$, $\boldsymbol{L}_2 \in \mathbb{R}^{300 \times 20}$, $\boldsymbol{L}_3 \in \mathbb{R}^{300 \times 10}$, $\boldsymbol{L}_4 \in \mathbb{R}^{300 \times 30}$, $\boldsymbol{L}_5 \in \mathbb{R}^{300 \times 20}$, and that $\boldsymbol{S} = \bigoplus_{i=1}^5 \boldsymbol{S}_i$ where $\boldsymbol{S}_1 \in \mathbb{R}^{20 \times 30}$, $\boldsymbol{S}_2 \in \mathbb{R}^{20 \times 30}$, $\boldsymbol{S}_3 \in \mathbb{R}^{10 \times 15}$, $\boldsymbol{S}_4 \in \mathbb{R}^{30 \times 45}$, $\boldsymbol{S}_5 \in \mathbb{R}^{20 \times 30}$. For the $i$-th group, the elements in $\boldsymbol{L}_i$ is sampled i.i.d from $\mathcal{N}(m_i, 0.01)$, where $m_i = 5i$. $\boldsymbol{S}_i$ is generated as $\boldsymbol{S}_i = s_i \mathbf{1}$, i.e., every element in $\boldsymbol{S}_i$ shares the same value. Moreover, $s_i$ is calculated from the constraint that $\boldsymbol{S} \in \Pi(\mathbf{a}, \mathbf{b})$. Then the task parameter is generated as $\boldsymbol{W} = \boldsymbol{LS}$. For each task, the outputs are generated as $\boldsymbol{Y}^{(i)} = \boldsymbol{X}^{(i)}(\boldsymbol{W}^{(i)} + \epsilon^{(i)})$, where $\epsilon^{(i)} \in \mathbb{R}^{200 \times 1}$, and $\epsilon^{(i)} \sim \mathcal{N}(0, 0.1^2 \boldsymbol{I}_{500})$. Based on this setting, we compare GBDSP with GOMTL in the simulation dataset to see how the block-diagonal structure benefits the latent task representation based MTL.

First, we show how well could GOMTL and GBDSP recover the block-diagonal structure. We compare $\boldsymbol{S}$ obtained from GOMTL and $\boldsymbol{S}^{\ddagger}$ obtained from GBDSP, with the initial value of $\hat{\boldsymbol{L}}$ set as $\hat{\boldsymbol{L}} = \boldsymbol{L} + \mathcal{N}(0, 0.05\boldsymbol{I})$. As shown in Fig.1(a)- Fig.1(c), GBDSP recovers a much clearer structure than GOMTL. Moreover, we provide a closer look at the embedding vectors in GBDSP. To do this, we visualize the spectral embeddings $\boldsymbol{f}$ in a 3d space with t-SNE [Maaten and Hinton, 2008], which is shown in Fig.2(c). In this figure, the points with different colors represent latent/output tasks in different groups. Clearly, we see that the clusters are well-separated in the spectral embedding space, which again verifies the grouping power of the proposed method.

Next, we check whether GBDSP could improve the performance with a structural LATM. In Fig.2, we plot the performance of GOMTL and GBDSP with different training set ratio

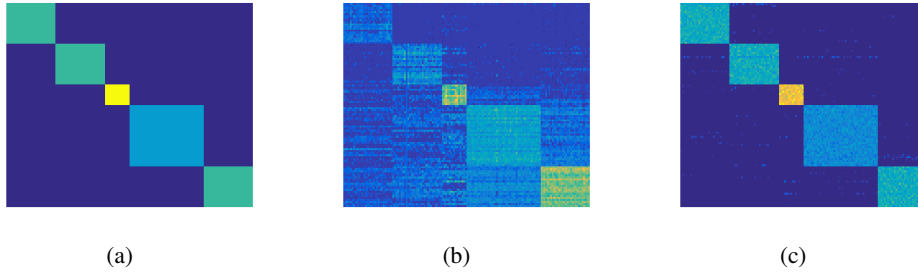

|(a)|(b)|(c)|

Figure 1: Visualizations over the Simulated Dataset. (a)-(c) provide structural comparisons over the LATM: (a) shows The true LATM; (b) shows the LATM recovered by GOMTL (c) shows the LATM recovered by GBDSP.

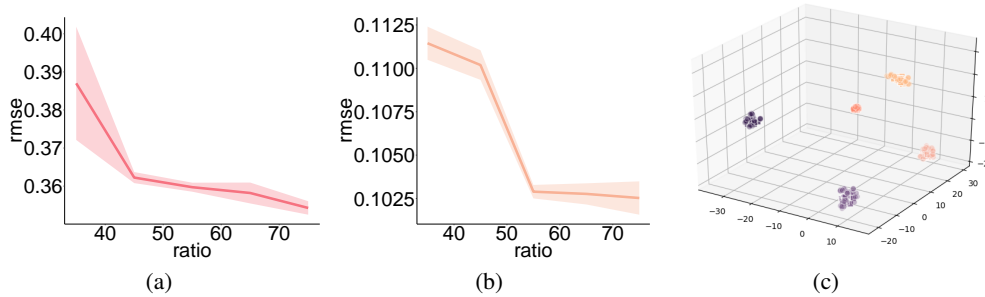

|(a)|(b)|(c)|

Figure 2: (a-b) Performance curve with different training data ratio: (a) GOMTL (b) GBDSP(c) shows the spectral group embedding of $o$ and $l$ in GBDSP.

$(0.35, 0.45, 0.55, 0.65, 0.75)$. The corresponding results show that GBDSP consistently provides a better performance with a smaller variance, which supports the idea that learning a block-diagonal LATM structure improves the performance.

## 4 Conclusion

To simultaneously leverage a latent task representation and alleviate the inter-group negative transfer issue, we develop a novel MTL method GBDSP, which simultaneously separates the latent tasks and out tasks into a given number of groups. Moreover, we adopt an optimization method to solve the model parameters, which gives an alternative update scheme for our multi-convex objective function. The solution produced by the optimization method shows a close connection between our method and the optimal transport problem, which brings new insight into how negative transfer could be prevented across latent tasks and output tasks. Furthermore, we provide theoretical analysis on the spectral properties of the model parameters. Empirical results on the simulated dataset show that GBDSP could roughly recover the correct grouping structure with good performance, and results on the real-world datasets further verify the effectiveness of our proposed model on the problem of personalized attribute prediction.

## 5 Acknowledgements

This work was supported in part by National Natural Science Foundation of China: 61620106009, U1636214, 61836002, 61861166002, 61672514 and 61976202, in part by National Basic Research Program of China (973 Program): 2015CB351800, in part by Key Research Program of Frontier Sciences, CAS: QYZDJ-SSW-SYS013, in part by the Strategic Priority Research Program of Chinese Academy of Sciences, Grant No. XDB28000000, in part by the Science and Technology Development Fund of Macau SAR (File no. 0001/2018/AFJ) Joint Scientific Research Project, in part by Beijing Natural Science Foundation (No. 61971016, L182057, and 4182079), in part by Peng Cheng

Laboratory Project of Guangdong Province PCL2018KP004, and in part by Youth Innovation Promotion Association CAS.

## Footnotes

[1] If $K$ is not the maximum of such numbers, we can always find out more disjoint blocks.

[2] Let us note that, in the rest of the paper, $\boldsymbol{S} = T\boldsymbol{S}^\ddagger$ no longer holds

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
