[Supplementary Material]

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

_1(\boldsymbol{A}), \cdots, \lambda_m(\boldsymbol{A})$ such that $\lambda_1(\boldsymbol{A}) \geq \lambda_2(\boldsymbol{A}) \geq \cdots \geq \lambda_m(\boldsymbol{A})$. $\langle \cdot, \cdot \rangle$ denotes the inner product for two matrices or two vectors. Given two vectors $\boldsymbol{a}$ and $\boldsymbol{b}$, $\boldsymbol{a} \oplus \boldsymbol{b}$ denotes the outer sum $\boldsymbol{a}\mathbf{1}^\top + \mathbf{1}\boldsymbol{b}^\top$.

Given two matrices $\boldsymbol{A}$ and $\boldsymbol{B}$, $\boldsymbol{A} \oplus \boldsymbol{B}$ denotes the direct sum of two matrices, i.e., $\boldsymbol{A} \oplus \boldsymbol{B} = \begin{bmatrix} \boldsymbol{A} & \boldsymbol{0} \\ \boldsymbol{0} & \boldsymbol{B} \end{bmatrix}$,

and we say $\boldsymbol{A} \succeq \boldsymbol{B}$, if $\boldsymbol{A} - \boldsymbol{B}$ is positive semi-definite. For distributions, $\mathcal{N}(\mu, \sigma^2)$ denotes the normal distribution. $\mathcal{P}_m$ denotes the set of all permutation matrices in $\mathbb{R}^{m \times m}$. For two matrices $\boldsymbol{A}$ and $\boldsymbol{B}$ having the same size, $d(\boldsymbol{A}, \boldsymbol{B}) = \|\boldsymbol{A} - \boldsymbol{

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

}) = \left\{ \left\{ \hat{Y}^{(i)}(\boldsymbol{X}_i^{(t)}) = (\boldsymbol{L}\boldsymbol{S}^{(i)})^{\top} \boldsymbol{X}_i^{(t)} \right\}_{ti} : \|L\|_F^2 \leq \xi_1, \right.$$

$$\left. d(\boldsymbol{S}, T\boldsymbol{S}^{\ddagger}) \leq \xi_2, \ \langle \Delta(\boldsymbol{S}^{\ddagger}), \boldsymbol{U} \rangle \leq \xi_3, \boldsymbol{S}^{\ddagger} \in \Pi(\mathbf{a}, \mathbf{b}), \boldsymbol{U} \in \mathcal{M} \right\},$$

where $\xi_1 = 2\mathcal{J}_0/\alpha_1, \xi_2 = 2\mathcal{J}_0/\alpha_2, \xi_3 = 2\mathcal{J}_0/\alpha_3$. Now we are ready to represent the theoretical results based on $\mathcal{H}$.

As the first step, we explore how well a model learned from $\mathcal{H}$ generalizes to the overall population. The empirical risk $\hat{\mathcal{R}}(\boldsymbol{L}, \boldsymbol{S})$ over the observed dataset and the task-averaged risk $\mathcal{R}(\boldsymbol{L}, \boldsymbol{S})$ are given as: $\hat{\mathcal{R}}(\boldsymbol{L}, \boldsymbol{S}) = \sum_{i=1}^{T} \mathcal{J}^{(i)}/T, \mathcal{R}(\boldsymbol{L}, \boldsymbol{S}) = \sum_{i=1}^{T} \mathbb{E}_{\mu_i}\left[\mathcal{J}^{(i)}\right]/T$, where the training data $\boldsymbol{X}_j^{(i)}, \boldsymbol{Y}_j^{(i)}$ are sampled from $\mu_i$. We then bound the term $\Delta = \mathcal{R}(\boldsymbol{L}, \boldsymbol{S}) - \hat{\mathcal{R}}(\boldsymbol{L}, \boldsymbol{S})$. Following the spirit of [Maurer et al., 2016], we have the following bound for the hypothesis space:

**Theorem 3.** *Suppose that $n_1 = n_2 \cdots = n_T = n$, the loss function $\ell(y, \cdot) : \hat{y} \mapsto [0, M]$, $\ell(y, \cdot)$ is $M\phi$-Lipschitz continuous, and $\forall (\boldsymbol{L}, \boldsymbol{S})$ are chosen from $\mathcal{H}$, the following bound holds with possibility at least $1 - \delta$ :*

$$\frac{\Delta}{M} \leq \kappa_1 \phi \aleph \left( \frac{\xi_1 k \|\boldsymbol{COV}(\boldsymbol{X})\|_1}{nT} \right)^{1/2} + 2\kappa_2 \phi \aleph \cdot \left( \frac{\xi_1 \|\boldsymbol{COV}(\boldsymbol{X})\|_{\infty}}{n} \right)^{1/2} + \left( \frac{9 \ln (2/\delta)}{2nT} \right)^{1/2},$$

*where $\kappa_1$ and $\kappa_2$ are two universal constants, $\aleph = \sqrt{\xi_2} + 1$. $\boldsymbol{COV}(\boldsymbol{X})$ is the covariance operator defined as $\langle \boldsymbol{COV}(\boldsymbol{X})\boldsymbol{u}, \boldsymbol{v} \rangle = (1/nT) \cdot \sum_{ti} \left\langle \boldsymbol{u}, \boldsymbol{X}_i^{(t)} \right\rangle \left\langle \boldsymbol{X}_i^{(t)}, \boldsymbol{v} \right\rangle$.*

The proof can be found in Appendix B.3. With the sample complexity given, we narrow our focus to the problem that how $\xi_2, \xi_3$ benefit the hypothesis space. The following theorem shows that $\xi_2, \xi_3$ control the spectral properties of $\boldsymbol{S}$ and $\boldsymbol{S}^{\ddagger}$.

**Theorem 4** (Spectral Properties of $\boldsymbol{S}$). *Let $k \leq T$, define the SVD of $\boldsymbol{S}$ as $\boldsymbol{S} = \boldsymbol{P}\boldsymbol{\Lambda}\boldsymbol{Q}^{\top}$, where $\boldsymbol{P} = [\boldsymbol{p}_1, \cdots, \boldsymbol{p}_k], \boldsymbol{Q} = [\boldsymbol{q}_1, \cdots, \boldsymbol{q}_k]$ are left and right singular vectors respectively, $\boldsymbol{\Lambda} = diag(\sigma_i(\boldsymbol{S}))$ with $\sigma_1(\boldsymbol{S}) \geq \sigma_2(\boldsymbol{S}) \cdots \geq \sigma_k(\boldsymbol{S}) \geq 0$. The following properties hold for all $\boldsymbol{S} \in \mathcal{H}$:*

(a) *The bottom $K$ eigenvalues of the graph Laplacian induced by $\boldsymbol{S}$ is bounded by : $\sum_{i=N-K+1}^{N} \lambda_i(\Delta(\boldsymbol{S})) \leq T\xi_3 + \sqrt{\xi_2 K}(\sqrt{2} + \sqrt{k} + \sqrt{T})$, where $\Delta(\boldsymbol{S})$ is obtained from replacing $\boldsymbol{S}^{\ddagger}$ in $\Delta(\boldsymbol{S}^{\ddagger})$ with $\boldsymbol{S}$ .*

(b) *Define $\mathbb{M}(\boldsymbol{P}) = Span\{\boldsymbol{p}_1, \cdots \boldsymbol{p}_K\}$ and $\mathbb{M}(\boldsymbol{Q}) = Span\{\boldsymbol{q}_1, \cdots \boldsymbol{q}_K\}$, if $\xi_3 + \sqrt{\xi_2}/T < 1/T$ and $rank(\boldsymbol{S}) \geq K$, then we have:*

$$\frac{\sigma_1(\boldsymbol{S})}{\sigma_K(\boldsymbol{S})} = \max_{\substack{\boldsymbol{x}, \boldsymbol{y} \in \mathbb{M}(\boldsymbol{P}) \\ \|\boldsymbol{x}\|_2 = 1, \|\boldsymbol{y}\|_2 = 1}} \frac{\|\boldsymbol{S}\boldsymbol{x}\|_2}{\|\boldsymbol{S}\boldsymbol{y}\|_2} = \max_{\substack{\boldsymbol{x}, \boldsymbol{y} \in \mathbb{M}(\boldsymbol{Q}) \\ \|\boldsymbol{x}\|_2 = 1, \|\boldsymbol{y}\|_2 = 1}} \frac{\|\boldsymbol{S}^{\top}\boldsymbol{x}\|_2}{\|\boldsymbol{S}^{\top}\boldsymbol{

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

# Appendix

## A Proofs of the Propositions

### A.1 Proof of Proposition 1

*Proof.* Given the solution of the $U$ subproblem, the $S^\ddagger$ subproblem could be formulated as:

$$\min_{S^\ddagger \in \Pi(\mathbf{a},\mathbf{b})} \frac{\alpha_2}{2}||S - S^\ddagger{}_c||_F^2 + \alpha_3 \cdot \left\langle diag(\begin{bmatrix} 0 & S^\ddagger \\ S^{\ddagger\top} & 0 \end{bmatrix}\mathbf{1}) - \begin{bmatrix} 0 & S^\ddagger \\ S^{\ddagger\top} & 0 \end{bmatrix}, U \right\rangle.$$

With the fact that

$$\left\langle diag(\begin{bmatrix} 0 & S^\ddagger \\ S^{\ddagger\top} & 0 \end{bmatrix}\mathbf{1}) - \begin{bmatrix} 0 & S^\ddagger \\ S^{\ddagger\top} & 0 \end{bmatrix}, U \right\rangle = \left\langle diag(U)\mathbf{1}^\top - U, \begin{bmatrix} 0 & S^\ddagger \\ S^{\ddagger\top} & 0 \end{bmatrix} \right\rangle,$$

and simple scaling of the constants, we could reformulate the problem as:

$$\min_{S^\ddagger \in \Pi(\mathbf{a},\mathbf{b})} \frac{\vartheta}{2}||\bar{S} - S^\ddagger||_F^2 + \left\langle \Delta^{(1)} + \Delta^{(2)\top}, S^\ddagger \right\rangle.$$

where $\Delta = diag(U)\mathbf{1}^\top - U$, $\Delta^{(1)} = \Delta(1:k, (k+1):end)$, $\Delta^{(2)} = \Delta((k+1):end, 1:k)$.

Then the proof directly follows the fact that

$$\Delta_{ij}^{(1)} + \Delta_{ji}^{(2)} = U_{ii} + U_{k+j,k+j} - U_{i,k+j} - U_{k+j,i} = \|\mathbf{f}_i - \mathbf{f}_{k+j}\|_2^2.$$

$\square$

### A.2 Proof of Proposition 2

*Proof.* **(a)**: Since $\Pi(\mathbf{a},\mathbf{b})$ is bounded, the sequence $\{S^\ddagger{}_\vartheta\}_{\vartheta \to 0}$ must admit at least one convergent subsequence. Pick any such subsequence with a limit point $S^{\ddagger *}$, and pick any $S^\ddagger \in$ $\operatorname{argmin}_{S^\ddagger \in \Pi(\mathbf{a},\mathbf{b})} \left\langle D, S^\ddagger \right\rangle$, from the optimality of $S^\ddagger{}_\vartheta$ in this subsequence and the optimality of $S^\ddagger$, we have:

$$0 < \left\langle D, S^\ddagger{}_\vartheta - S^\ddagger \right\rangle \le \vartheta \cdot \left( d(S^\ddagger, \bar{S}) - d(S^\ddagger{}_\vartheta, S_c) \right).$$

Now we prove that $S^{\ddagger *}$ is a feasible solution. Obviously since $\Pi(\mathbf{a},\mathbf{b})$ is closed, we have $S^{\ddagger *} \in$ $\Pi(\mathbf{a},\mathbf{b})$. Moreover, taking the limit $\vartheta \to 0$ in the inequality above, we have $\left\langle D, S^{\ddagger *} - S^\ddagger \right\rangle = 0$. This implies that $S^{\ddagger *} \in \operatorname{argmin}_{S^\ddagger \in \Pi(\mathbf{a},\mathbf{b})} \left\langle D, S^\ddagger \right\rangle$. Above all, we know $S^{\ddagger *}$ is a feasible solution. Now we continue to show its optimality. Again, by dividing $\vartheta$ on both sides of the inequality above and taking the limit $\vartheta \to 0$, we have: $d(S^\ddagger, \bar{S}) - d(S^{\ddagger *}, \bar{S}) \ge 0$. This implies that $d(S^\ddagger, \bar{S})$ reaches a minimal value in the feasible set at $S^\ddagger = S^{\ddagger *}$. Moreover, since the optimization problem is strongly convex, we know such $S^{\ddagger *}$ must be unique. Since $\Pi(\mathbf{a},\mathbf{b})$ is closed and every convergent subsequence of $\{S^\ddagger{}_\vartheta\}$ converges to the optimal solution of the problem, $\{S^\ddagger{}_\vartheta\}$ converges to the optimal solution of the problem. This ends our proof.

**(b)** Let $S_1^\star$ and $S_2^\star$ be the (an) optimal solution of $\mathcal{J}_{REG}$ and $\mathcal{J}_{OT}$, respectively. From the optimality of $S_1^\star$, we have:

$$\langle D, S_1^\star \rangle + \frac{\vartheta}{2}\|S_1^\star - \bar{S}\|_F^2 \le \langle D, S_2^\star \rangle + \frac{\vartheta}{2}\|S_2^\star - \bar{S}\|_F^2.$$

Furthermore, according to the optimality of $S_2^\star$, we have:

$$\langle D, S_2^\star \rangle + \frac{\vartheta}{2}\|S_1^\star - \bar{S}\|_F^2 \le \langle D, S_1^\star \rangle + \frac{\vartheta}{2}\|S_1^\star - \bar{S}\|_F^2.$$

Above all we have:

$$\frac{\vartheta}{2}\|S_1^\star - \bar{S}\|_F^2 \le \mathcal{J}_{REG} - \mathcal{J}_{OT} \le \frac{\vartheta}{2}\|S_2^\star - \bar{S}\|_F^2.$$

Since $\boldsymbol{S}_1^\star \in \Pi(\mathbf{a}, \mathbf{b})$, we must have:

$$\min_{\boldsymbol{S} \in \Pi(\mathbf{a},\mathbf{b})} \|\boldsymbol{S} - \bar{\boldsymbol{S}}\|_F^2 \leq \frac{\vartheta}{2} \|\boldsymbol{S}_1^\star - \bar{\boldsymbol{S}}\|_F^2.$$

Moreover, we have:

$$\min_{\boldsymbol{S}\mathbf{1}=\mathbf{a}} \|\boldsymbol{S} - \bar{\boldsymbol{S}}\|_F^2 \leq \min_{\boldsymbol{S} \in \Pi(\mathbf{a},\mathbf{b})} \|\boldsymbol{S} - \bar{\boldsymbol{S}}\|_F^2, \qquad \min_{\boldsymbol{S}^\top\mathbf{1}=\mathbf{b}} \|\boldsymbol{S} - \bar{\boldsymbol{S}}\|_F^2 \leq \min_{\boldsymbol{S} \in \Pi(\mathbf{a},\mathbf{b})} \|\boldsymbol{S} - \bar{\boldsymbol{S}}\|_F^2.$$

It is easy to see that

$$\min_{\boldsymbol{S}\mathbf{1}=\mathbf{a}} \frac{1}{2}\|\boldsymbol{S} - \bar{\boldsymbol{S}}\|_F^2 = \frac{\|\bar{\boldsymbol{S}}\mathbf{1} - \mathbf{a}\|_2^2}{T}, \qquad \min_{\boldsymbol{S}^\top\mathbf{1}=\mathbf{b}} \frac{1}{2}\|\boldsymbol{S} - \bar{\boldsymbol{S}}\|_F^2 = \frac{\|\bar{\boldsymbol{S}}^\top\mathbf{1} - \mathbf{b}\|_2^2}{k}.$$

All these lead to:

$$\vartheta \cdot \max\left\{ \frac{\|\bar{\boldsymbol{S}}\mathbf{1} - \mathbf{a}\|_2^2}{T}, \frac{\|\bar{\boldsymbol{S}}^\top\mathbf{1} - \mathbf{b}\|_2^2}{k} \right\} \leq \mathcal{J}_{REG} - \mathcal{J}_{OT}.$$

Moreover, we have $\|\boldsymbol{S} - \bar{\boldsymbol{S}}\|_F^2 \leq 2\|\boldsymbol{S}\|_F^2 + 2\|\bar{\boldsymbol{S}}\|_F^2$. For all $\boldsymbol{S} \in \Pi(\mathbf{a}, \mathbf{b})$, we have:

$$\|\boldsymbol{S}\|_F^2 = \sum_{i,j} S_{i,j} = \sum_i a_i^2 \sum_j \left(\frac{S_{i,j}}{a_i}\right)^2 \leq \sum_i a_i^2 \sum_j \left(\frac{S_{i,j}}{a_i}\right) \leq \|\mathbf{a}\|_2^2.$$

Similarly, we have $\|\boldsymbol{S}\|_F^2 \leq \|\mathbf{b}\|_2^2$. These lead to :

$$\mathcal{J}_{REG} - \mathcal{J}_{OT} \leq \vartheta \cdot (\|\bar{\boldsymbol{S}}\|_F^2 + \min\{\|\mathbf{a}\|_2^2, \|\mathbf{b}\|_2^2\}).$$

$\square$

### A.3 Proof of Proposition 3

*Proof.* The lagrangian dual of the problem could be written as:

$$\max_{\boldsymbol{f},\boldsymbol{g},\boldsymbol{\Lambda} \geq 0} \min_{\boldsymbol{S}^\ddagger} \frac{\vartheta}{2}\|\bar{\boldsymbol{S}} - \boldsymbol{S}^\ddagger\|_F^2 + \left\langle \mathcal{D}, \boldsymbol{S}^\ddagger \right\rangle - \left\langle \boldsymbol{S}^\ddagger\mathbf{1} - \mathbf{a}, \boldsymbol{f} \right\rangle - \left\langle \boldsymbol{S}^{\ddagger^\top}\mathbf{1} - \mathbf{b}, \boldsymbol{g} \right\rangle - \left\langle \boldsymbol{\Lambda}, \boldsymbol{S}^\ddagger \right\rangle. \quad (1)$$

Since ($Primal$) is strongly convex, the strong duality reduces to the Slater condition. It is easy to see that ($Primal$) satisfies this condition, since $\boldsymbol{S}_\dagger = \frac{\mathbf{1}_{k \times T}}{kT} \in \Pi(\mathbf{a}, \mathbf{b})$ and obviously $\boldsymbol{S}_\dagger > 0$. The primal problem could then be solved from its dual problem Eq.(1). Now we show that it is equivalent to solving ($Dual$). Solving the inner minimization problem of Eq.(1), we have:

$$\boldsymbol{S}^\ddagger = \frac{\boldsymbol{h} \oplus \boldsymbol{g} + \boldsymbol{\Lambda} - \mathcal{D}}{\vartheta} + \bar{\boldsymbol{S}}. \quad (2)$$

Note that since the inner minimization problem is strongly convex toward $\boldsymbol{S}^\ddagger$, the solution is unique. Plugging the solution into the outer maximization problem yields:

$$\max_{\boldsymbol{f},\boldsymbol{g},\boldsymbol{\Lambda} \geq 0} \frac{1}{2\vartheta}\|\boldsymbol{h} \oplus \boldsymbol{g} + \boldsymbol{\Lambda} - \mathcal{D}\|_F^2 + \langle \boldsymbol{f}, \mathbf{a} \rangle + \langle \boldsymbol{g}, \mathbf{b} \rangle + \left\langle \mathcal{D} - \boldsymbol{h} \oplus \boldsymbol{g} - \boldsymbol{\Lambda}, \frac{\boldsymbol{h} \oplus \boldsymbol{g} + \boldsymbol{\Lambda} - \mathcal{D}}{\vartheta} + \bar{\boldsymbol{S}} \right\rangle.$$

Fixing $\boldsymbol{f}, \boldsymbol{g}$, we see the maximization problem is strongly concave toward $\boldsymbol{\Lambda}$. This means that $\boldsymbol{\Lambda}$ can be uniquely determined by $\boldsymbol{f}, \boldsymbol{g}$ with the solution of the partial maximization problem:

$$\max_{\boldsymbol{\Lambda} \geq 0} -\frac{1}{2\vartheta}\|\boldsymbol{h} \oplus \boldsymbol{g} + \boldsymbol{\Lambda} - \mathcal{D} + \vartheta\bar{\boldsymbol{S}}\|_F^2 + \langle \boldsymbol{f}, \mathbf{a} \rangle + \langle \boldsymbol{g}, \mathbf{b} \rangle + const. \quad (3)$$

This yields:

$$\boldsymbol{\Lambda} = \left[\boldsymbol{h} \oplus \boldsymbol{g} + \boldsymbol{\Lambda} - \mathcal{D} + \bar{\boldsymbol{S}}\right]_-. \quad (4)$$

Plugging Eq.(4) into Eq.(2) and (3) then complete the proof. $\square$

# B Proofs of the Theorems

## B.1 Proof of Theorem 1

*Proof.*

**(a)** We first show that if $\chi_{\boldsymbol{S}^\ddagger} = K$ then $\dim(Null(\Delta(\boldsymbol{S}^\ddagger))) = K$. Since $\chi_{\boldsymbol{S}^\ddagger} = K$, $\exists \boldsymbol{P}_r \in \mathcal{P}_k$, $\exists \boldsymbol{P}_c \in \mathcal{P}_T$, such that: $\boldsymbol{P}_r \boldsymbol{S}^\ddagger \boldsymbol{P}_C = \bigoplus_{i=1}^{K} \hat{\boldsymbol{S}}^{(i)}$, where $\hat{\boldsymbol{S}}^{(i)} \in \mathbb{R}^{k_i \times T_i}$, $\hat{\boldsymbol{S}}^{(i)} \neq \boldsymbol{0}$, $\forall i = 1, 2, \cdots, K$.
Let $\widetilde{\boldsymbol{P}}_0 = \begin{bmatrix} \boldsymbol{P}_r, 0 \\ 0, \boldsymbol{P}_c^\top \end{bmatrix}$, we have $\widetilde{\boldsymbol{P}}_0 \in \mathcal{P}_{k+T}$ and :

$$\widetilde{\boldsymbol{A}} = \widetilde{\boldsymbol{P}}_0 \boldsymbol{A}_{l \cup o} \widetilde{\boldsymbol{P}}_0^\top = \begin{bmatrix} 0 & \boldsymbol{K}_1 \\ \boldsymbol{K}_2 & 0 \end{bmatrix},$$

where $\boldsymbol{K}_1 = \bigoplus_{i=1}^{K} \hat{\boldsymbol{S}}^{(i)}$, $\boldsymbol{K}_2 = \bigoplus_{i=1}^{K} \hat{\boldsymbol{S}}^{(i)^\top}$. In the following, we show that $\widetilde{\boldsymbol{A}}$ could be rearranged to a direct sum of $K$ submatrices, following the same permutation on its rows and columns. We denote $\boldsymbol{e}_i = [e_{i,1} \cdots e_{i,k+T}]^\top$, where $e_{i,j} = 1$, $if\ i = j$, otherwise we have $e_{i,j} = 0$. Furthermore, we define $\boldsymbol{e}_{a:b} = [\boldsymbol{e}_a, \boldsymbol{e}_{a+1}, \cdots \boldsymbol{e}_b]$. Define a permutation matrix

$$\widetilde{\boldsymbol{P}}_1 = \left[ \boldsymbol{e}_{1:k_1}, \boldsymbol{e}_{(k+1):(k+T_1)}, \boldsymbol{e}_{(k_1+1):d_2}, \boldsymbol{e}_{(k+T_1+1):(k+T_2)}, \cdots, \boldsymbol{e}_{(d-k_K+1):d}, \boldsymbol{e}_{(k+T-t_K+1):(k+T)} \right]$$

Then we have:

$$\widetilde{\boldsymbol{A}}_1 = \widetilde{\boldsymbol{P}}_1^\top \widetilde{\boldsymbol{A}} \widetilde{\boldsymbol{P}}_1 = \bigoplus_{i=1}^{K} \boldsymbol{E}_i, \ \boldsymbol{E}_i = \begin{bmatrix} \boldsymbol{0} & \hat{\boldsymbol{S}}^{(i)} \\ \hat{\boldsymbol{S}}^{(i)\top} & \boldsymbol{0} \end{bmatrix}$$

Since $\chi_{\boldsymbol{S}^\ddagger}(\boldsymbol{P}_r, \boldsymbol{P}_c)$ reaches its maximum at K, the corresponding Graph $\widetilde{\mathcal{G}}$ with $\widetilde{\boldsymbol{A}}_1$ being its affinity matrix has $K$ connected components. Meanwhile, $\widetilde{\boldsymbol{A}}_1$ differs from $\boldsymbol{A}_{l \cup o}$ only by row and column permutations. This means that $\widetilde{\mathcal{G}}$ only rearranges the nodes in $\mathcal{G}_{l \cup o}$. Hence, $\mathcal{G}_{l \cup o}$ also has $K$ connected bipartite components. From the spectral property of a graph Laplacian [Von Luxburg, 2007], we have $\dim(Null(\Delta(\boldsymbol{S}))) = K$. This ends the proof of **(a)**.

**(b)** Next, we show that if $\dim(Null(\Delta(\boldsymbol{S}))) = K$ then $\chi_{\boldsymbol{S}^\ddagger} = K$. Since $\dim(Null(\Delta(\boldsymbol{S}))) = K$ , we know that $\mathcal{G}_{l \cup o}$ has $K$ connected components $g_1, \cdots g_k$. Denote $g(\boldsymbol{l}_i)$ as the corresponding group index that $\boldsymbol{l}_i$ belongs to and $g(\boldsymbol{o}_j)$ be the group index that $\boldsymbol{o}_j$ belongs to. Furthermore let $\mathcal{G}_l$ be a rearrangement of the indexes of latent tasks such that if $g(\boldsymbol{l}_{i_1}) < g(\boldsymbol{l}_{i_2})$ then $\mathcal{G}_l(i_1) < \mathcal{G}_l(i_2)$, if $g(\boldsymbol{l}_{i_1}) = g(\boldsymbol{l}_{i_2})$, then $\mathcal{G}_l(i_1) < \mathcal{G}_l(i_2)$ only if $i_1 < i_2$. Meanwhile, let $\mathcal{G}_o$ be the same rearrangement of the indexes of output tasks. Then define two permutation matrix $\boldsymbol{P}_r = [\boldsymbol{e}_{\mathcal{G}_l(1)}^{k^\top}; \boldsymbol{e}_{\mathcal{G}_l(2)}^{k^\top}; \cdots; \boldsymbol{e}_{\mathcal{G}_l(k)}^{k^\top}]$ and $\boldsymbol{P}_c = [\boldsymbol{e}_{\mathcal{G}_o(1)}^T, \boldsymbol{e}_{\mathcal{G}_o(2)}^T, \cdots, \boldsymbol{e}_{\mathcal{G}_o(k)}^T]$, where $\boldsymbol{e}_j^i \in \mathbb{R}^i$, $\boldsymbol{e}_{j,k}^i = 0$ if $j \neq k$, $\boldsymbol{e}_{j,j}^i = 1$. Then we have $\boldsymbol{P}_r \boldsymbol{S}^\ddagger \boldsymbol{P}_c = \bigoplus \hat{\boldsymbol{S}}^{(i)}$. Now we prove that $\widehat{\boldsymbol{S}}^{(i)} \neq \boldsymbol{0}, \forall\ i$ by contradiction. Without loss of generality, we assume that $\widehat{\boldsymbol{S}}^{(K)} = \boldsymbol{0}$, then all the elements in the last $k_i$ rows and the last $T_i$ columns of $\boldsymbol{P}_r \boldsymbol{S}^\ddagger \boldsymbol{P}_C$ must be 0. This contradicts with the fact that $\boldsymbol{P}_r \boldsymbol{S}^\ddagger \boldsymbol{P}_c \boldsymbol{1} = \boldsymbol{P}_r \boldsymbol{S}^\ddagger \boldsymbol{1} > \boldsymbol{0}$ and $\boldsymbol{P}_c^\top \boldsymbol{S}^{\ddagger^\top} \boldsymbol{P}_r^\top \boldsymbol{1} = \boldsymbol{P}_c^\top \boldsymbol{S}^{\ddagger^\top} \boldsymbol{1} > \boldsymbol{0}$. This implies that $\chi_{\boldsymbol{S}^\ddagger} \geq K$. Then we show that $\chi_{\boldsymbol{S}^\ddagger}$ could not exceed K by contradiction. If $\chi_{\boldsymbol{S}^\ddagger} > K$, then at least one $\hat{\boldsymbol{S}}^{(i)}$ could be written as a direct sum of two nonzero sub-matrices. This contradicts with the fact that $\begin{bmatrix} \boldsymbol{0} & \hat{\boldsymbol{S}}^{(i)} \\ \hat{\boldsymbol{S}}^{(i)\top} & \boldsymbol{0} \end{bmatrix}$ is a connected bipartite component. This shows that $\chi_{\boldsymbol{S}^\ddagger} \leq K$. Above all, we must have $\chi_{\boldsymbol{S}^\ddagger} = K$.

$\square$

## B.2 Proof of Theorem 2

We note that the proof is similar to the literature [Overton and Womersley, 1992a, Alizadeh, 1995], where the largest $K$ eigenvalues instead of the smallest eigenvalues are analyzed. We provide a proof here to make our paper self-contained.

*Proof.* Denote the eigenvalue decomposition of $\boldsymbol{A}$ as

$$\boldsymbol{A} = \boldsymbol{Q} \Lambda \boldsymbol{Q}^\top, \ \Lambda = diag(\lambda_1(\boldsymbol{A}), \cdots, \lambda_N(\boldsymbol{A})).$$

For any element $U$ in the feasible set $\Gamma$, we have: $\langle A, U \rangle = \sum_i C_{ii}\lambda_i(A)$, where $C = Q^\top U Q$. Since $C$ has the same eigenvalues as $U$, we have $C \in \Gamma$ if and only if $U \in \Gamma$. Then we have:

$$\min_{U \in \Gamma} \langle A, U \rangle \iff \min_{C \in \Gamma} \sum_i C_{ii}\lambda_i(A). \tag{5}$$

Define $e^i \in \mathbb{R}^{N \times 1}$, $e_i^i = 1$ and $e_s^i = 0, if\ s \neq i$, then we reach the fact that: $C_{ii} = \dfrac{e^{i\top} C e^i}{e^{i\top} e^i}$. We could then attain the following inequality based on the extremal property of the top/bottom eigenvalue of $C$:

$$0 \leq \lambda_N(C) = min_{x}\frac{x^\top C x}{x^\top x} \leq C_{ii} \leq max_{x}\frac{x^\top C x}{x^\top x} = \lambda_1(C) \leq 1.$$

Since $\lambda_1(A) \geq \lambda_2(A) \cdots \geq \lambda_N(A)$, the minimum of (5) is reached at $\sum_{i=N-K+1}^{N} \lambda_1(A)$ when $C_{ii} = 0, i \leq N - K$, $C_{ii} = 1, i \geq N - k + 1$. This directly shows that $\sum_{N-K+1}^{N} \lambda_i(A) = \min_{U \in \Gamma} \langle A,\ U \rangle$.

Now it only remains to prove that $U = V_K V_K^\top$ is an optimal solution. Since $V_K$ is the eigenvectors associated with the smallest $K$ eigenvalues of $A$, we have $Q = [V_K^\perp, V_K]$, where $V_K^\perp$ denotes the eigenvectors associated with the largest $N - K$ eigenvalues, and we have $V_K^\top V_K^\perp = 0$ and $V_K^{\perp \top} V_K = 0$. In this sense, we obtain:

$$Q^\top U Q = \begin{bmatrix} V_K^{\perp \top} \\ V_K^\top \end{bmatrix} V_K V_K^\top [V_K^\perp, V_K] = \begin{bmatrix} 0 \\ I_K \end{bmatrix} [0, I_K] = \begin{bmatrix} 0 & 0 \\ 0 & I_K \end{bmatrix}.$$

Then the proof follows that $C = Q^\top U Q$ satisfies the optimal condition analysed above. $\qquad \square$

### B.3   Proof of Theorem 3

In the proof of Thm.3, we adopt the Gaussian Average as the complexity measure of the hypothesis space, which is defined as follows.

**Definition 2** (Gaussian Average). *Given a set $\mathcal{C} \subset \mathbb{R}^n$, the Gaussian Average is defined as*

$$\mathbb{G}(\mathcal{C}) = \mathbb{E}\left[\sup_{c \in \mathcal{C}} \langle \gamma, c \rangle\right],$$

*where $\gamma = [\gamma_1 \cdots \gamma_n]$ and $\gamma_1 \cdots \gamma_n \overset{i.i.d}{\sim} \mathcal{N}(0, 1)$.*

Define the multi-task dataset over T tasks $\mathcal{X} = \{X^{(i)}\}_{i=1}^T$, where $X^{(i)} = \{X_1^{(i)}, \cdots, X_n^{(i)}\}$. Given an $M\phi$-Lipschitz continuous loss function $\ell$ with range $[0, M]$, we are interested in the following classes.

**1) Target hypothesis Space**. Recall we define the target hypothesis space as $\mathcal{H}(L, S, S^\ddagger, U)$. For the case of simplicity, we rename it here as $\mathcal{F}(\mathcal{X})$.

$$\mathcal{F}(\mathcal{X}) = \mathcal{H}(L, S, S^\ddagger, U) = \Bigg\{ \left\{ \hat{Y}^{(i)}(X_i^{(t)}) = (LS^{(i)})^\top X_i^{(t)} \right\}_{ti} :$$
$$\|L\|_F^2 \leq \xi_1,\ d(S, TS^\ddagger) \leq \xi_2,\ \langle \Delta(S^\ddagger), U \rangle \leq \xi_3, S^\ddagger \in \Pi(\mathbf{a}, \mathbf{b}), U \in \mathcal{M} \Bigg\}.$$

To bound $\Delta$, we need to analysis $\ell \circ \mathcal{F}(\mathcal{X})$, which is formed as the composition of the loss function and the hypothesis space:

$$\ell \circ \mathcal{F}(\mathcal{X}) = \left\{ \{\{l(f_t(X_i^{(t)}), Y_i^{(t)})\}_{i=1}^n\}_{t=1}^T : \left\{ f_t(X_i^{(t)}) \right\}_{ti} \in \mathcal{F}(\mathcal{X}) \right\}$$

**2) Latent response space**. Moreover, given the latent task embedding $L = [L^{(1)}, \cdots L^{(k)}]$, then $L^{(j)} X_i^{(t)}$ gives a response for $X_i^{(t)}$ on the $j$-th latent task. With the choice of $\mathcal{F}(\mathcal{X})$, one could only choose latent responses from the following space.

$$\mathcal{L}_c(\mathcal{X}) = \left\{ \{L^{(k)\top} X_i^{(t)}\}_{kti} : L = [L^{(1)}, \cdots L^{(k)}], \|L\|_F \leq c \right\}.$$

**3) Output response space**. We define the output response class for a single task as :

$$\mathcal{I}_c = \left\{ \boldsymbol{y} \in \mathbb{R}^k \mapsto \sum_{i=1}^{k} y_i \cdot s_i : \boldsymbol{s} = [s_1 \cdots s_k]^\top, ||\boldsymbol{s}||_2 \le c \right\}.$$

For $\mathcal{I}_c$, we define two quantities that are essential for the proof:

$$Q(\mathcal{I}_c) = \sup_{\boldsymbol{y} \ne \boldsymbol{y}' \in \mathbb{R}^{kn}} \frac{1}{||\boldsymbol{y} - \boldsymbol{y}'||} \mathbb{E} \left[ \sup_{s \in \mathcal{I}_c} \sum_{i=1}^{n} \gamma_i \left( s(\boldsymbol{y}_i) - s(\boldsymbol{y}_i') \right) \right]$$

where $\boldsymbol{y}$ concatenates all the $k$-dimensional latent responses for all samples (n) in a single task, i.e.,
$\boldsymbol{y} = [\boldsymbol{y}_1; \cdots ; \boldsymbol{y}_n], \boldsymbol{y}' = [\boldsymbol{y}_1'; \cdots ; \boldsymbol{y}_n'], \boldsymbol{y}_i, \boldsymbol{y}_i' \in \mathbb{R}^{k \times 1}$ are responses for the $i$-th instance. Moreover, we define the supremum of the Lipschitz constants over $\mathcal{I}_c$ as

$$Lip(\mathcal{I}_c) = \sup \{ B : ||s(\boldsymbol{x}) - s(\boldsymbol{x}')|| \le B||\boldsymbol{x} - \boldsymbol{x}'||, s \in \mathcal{I}_c \}$$

Given $\mathcal{I}_c$, we define the representation space as:

$$\mathcal{S}_c = \left\{ \boldsymbol{y} \in \mathbb{R}^{kTn} \mapsto \{f^{(t)}(\boldsymbol{y}_{ti})\}_{ti} : f^{(t)}(\cdot) \in \mathcal{I}_c \right\}$$

Here $\boldsymbol{y}, \boldsymbol{y}'$ concatecate all latent responses for all tasks and all samples, i.e., $\boldsymbol{y} = [\boldsymbol{y}_{11}; \cdots ; \boldsymbol{y}_{1n}; \cdots ; \boldsymbol{y}_{T1}; \cdots ; \boldsymbol{y}_{Tn}]$, where $\boldsymbol{y}_{ti}, \boldsymbol{y}_{ti}'$ could be regarded as latent responses for the $i$-th instance of the $t$-th task.

The following Lemma shows that the generalization ability could be controlled by $\mathbb{G}(\ell \circ \mathcal{F}(\mathcal{X}))$.

**Lemma 1.** *[Thm.9 of [Maurer et al., 2016]] Suppose that $n_1 = n_2 \cdots = n_T = n$, the loss function $l(y, \cdot) : \hat{y} \mapsto [0, M], \forall (\boldsymbol{L}, \boldsymbol{S})$ chosen from $\mathcal{F}(\mathcal{X}) = \mathcal{H}(\boldsymbol{L}, \boldsymbol{S}, \boldsymbol{S}^\ddagger, \boldsymbol{U})$ the following bound holds with possibility at least $1 - \delta$ :*

$$\frac{\Delta}{M} \le \frac{\sqrt{2\pi} \mathbb{G} \left( \frac{\ell \circ \mathcal{F}(\mathcal{X})}{M} \right)}{nT} + \sqrt{\frac{9ln(2/\delta)}{2nT}}.$$

Then we proceed the derivation by providing an explicit bound for $\mathbb{G}(\ell \circ \mathcal{F}(\mathcal{X}))$. The following Lemma shows that $\mathbb{G}(\ell \circ \mathcal{F}(\mathcal{X}))$ could be bounded above by the Gaussian Average of a simpler class.

**Lemma 2.** *If $\ell(\cdot, y)$ is $M\phi$-Lipschitz continuous, we have:*

$$\mathbb{G} \left( \frac{\ell \circ \mathcal{F}(\mathcal{X})}{M} \right) \le \phi \cdot \mathbb{G}(\mathcal{S}_\aleph \circ \mathcal{L}_{\xi_1^{1/2}}(\mathcal{X})),$$

*where $\aleph = \sqrt{\xi_2} + 1$.*

*Proof.* Since $\ell$ is $\phi$-Lipschitz continuous, we have:

$$\mathbb{G} \left( \frac{\ell \circ \mathcal{F}(\mathcal{X})}{M} \right) \le \phi \cdot \mathbb{G}(\mathcal{F}(\mathcal{X})).$$

For every $\left\{ \hat{Y}^{(t)}(\boldsymbol{x}_i^{(t)}) \right\}_{ti} \in \mathcal{H}(\boldsymbol{L}, \boldsymbol{S}, \boldsymbol{S}^\ddagger, \boldsymbol{U})$, the corresponding $\boldsymbol{L}$ and $\boldsymbol{S}^{(i)}$ satisfy : $||\boldsymbol{L}||_F \le \sqrt{\xi_1}$
and

$$||\boldsymbol{S}^{(i)}||_2 \le \sqrt{\xi_2} + ||T\boldsymbol{S}^{\ddagger(i)}||_2 \le \sqrt{\xi_2} + ||T\boldsymbol{S}^{\ddagger(i)}||_2 \le \sqrt{\xi_2} + \frac{T}{T} = \sqrt{\xi_2} + 1.$$

This implies that $\mathcal{F}(\mathcal{X}) \subset \mathcal{S}_\aleph \circ \mathcal{L}_{\xi_1^{1/2}}(\mathcal{X})$. According to the definition of Gaussian Average, we then reach:

$$\mathbb{G}(\mathcal{F}(\mathcal{X})) \le \mathbb{G}(\mathcal{S}_\aleph \circ \mathcal{L}_{\xi_1^{1/2}}(\mathcal{X})).$$

$\square$

**Lemma 3.**

$$\mathbb{G}(\mathcal{S}_\aleph \circ \mathcal{L}_{\xi_1^{1/2}}(\mathcal{X})) \le \kappa_1 \aleph \cdot \left( \xi_1 nkT ||\boldsymbol{COV}(\boldsymbol{X})||_1 \right)^{1/2} + 2\kappa_2 \aleph \cdot T \left( \xi_1 nk ||\boldsymbol{COV}(\boldsymbol{X})||_\infty \right)^{1/2}.$$

*Proof.* From the composition rule of Gaussian Average [Maurer et al., 2016], we have:

$$\mathbb{G}(\mathcal{S}_{\aleph} \circ \mathcal{L}_{\xi_1^{1/2}}(\mathcal{X})) \leq \kappa_1 \cdot Lip(\mathcal{I}_{\aleph}) \cdot \mathbb{G}(\mathcal{L}(\mathcal{X})) + \kappa_2\sqrt{T} \cdot Diam[\mathcal{L}(\mathcal{X})] \cdot Q(\mathcal{I}_{\aleph}),$$

where $Diam[\mathcal{L}(\mathcal{X})] = \sup_{\boldsymbol{y},\boldsymbol{y}' \in \mathcal{L}(\mathcal{X})} ||\boldsymbol{y} - \boldsymbol{y}'||$. Then we bound $Lip(\mathcal{I}_{\aleph})$, $\mathbb{G}(\mathcal{L}(\mathcal{X}))$, $Diam(\mathcal{L}(\mathcal{X}))$, and $Q(\mathcal{I}_{\aleph})$, respectively.

- $\forall f \in \mathcal{I}_c$, uniformly we have:

$$|f(\boldsymbol{x}_1) - f(\boldsymbol{x}_2)| \leq \aleph\|\boldsymbol{x}_1 - \boldsymbol{x}_2\|, \tag{6}$$

  since $f$ is a linear functional induced by a vector $\boldsymbol{s}$ with $||\boldsymbol{s}|| \leq \aleph$. This suggests that $Lip(\mathcal{I}_{\aleph}) \leq \aleph$.

- For $\mathbb{G}(\mathcal{L}(\mathcal{X}))$, we have:

$$\mathbb{G}(\mathcal{L}(\mathcal{X})) = \mathbb{E}\left[\sup_{\mathcal{L}(\mathcal{X})} \sum_{jtn} \gamma_{jtn} \cdot \left(\boldsymbol{L}^{(j)\top}\boldsymbol{X}_i^{(t)}\right)\right]$$

$$= \mathbb{E}\left[\sup_{\mathcal{L}(\mathcal{X})} \sum_{j} \boldsymbol{L}^{(j)\top}\left(\sum_{ti} \gamma_{jtn} \cdot \boldsymbol{X}_i^{(t)}\right)\right]$$

$$\leq \sqrt{\xi_1}\mathbb{E}\left[\left(\sum_{j}\left\|\sum_{ti}\gamma_{jti}\boldsymbol{x}_i^{(t)}\right\|^2\right)^{1/2}\right]$$

$$\leq \sqrt{\xi_1}\left(\sum_{j}\mathbb{E}\left[\left\|\sum_{ti}\gamma_{jti}\boldsymbol{x}_i^{(t)}\right\|^2\right]\right)^{1/2}$$

$$\leq \left(\xi_1 k \sum_{ti}\|\boldsymbol{x}_i^{(t)}\|^2\right)^{1/2} \leq \left(\xi_1 knT\|\boldsymbol{COV}(\boldsymbol{X})\|_1\right)^{1/2}.$$

- Since $Diam((\mathcal{L}(\mathcal{X}))) \leq 2\sup_{\boldsymbol{y} \in \mathcal{L}(\mathcal{X})} ||\boldsymbol{y}||$, we have:

$$Diam[\mathcal{L}(\mathcal{X})] \leq 2\sup_{||\boldsymbol{L}||_F \leq \xi_1^{1/2}}\left(\sum_{jti}(\boldsymbol{L}^{(j)\top}\boldsymbol{x}_i^{(t)})^2\right)^{1/2}$$

$$\leq 2\sup_{||\boldsymbol{L}||_F \leq \xi_1^{1/2}}\left(\sum_{j}\|\boldsymbol{L}^{(j)}\|_2^2 \cdot \sum_{ti}\left(\left(\frac{\boldsymbol{L}^{(j)}}{\|\boldsymbol{L}^{(j)}\|_2}\right)^{\top}\boldsymbol{x}_i^{(t)}\right)^2\right)^{1/2}$$

$$\leq 2\xi_1^{1/2} \cdot \left(k \cdot \sum_{ti}\sup_{\|\boldsymbol{y}\|_2=1}\left(\boldsymbol{y}^{\top}\boldsymbol{x}_i^{(t)}\right)^2\right)^{1/2}$$

$$\leq 2\left(\xi_1 nkT \cdot \|\boldsymbol{COV}(\boldsymbol{X})\|_{\infty}\right)^{1/2}.$$

- For $Q(\mathcal{I}_\aleph)$, we have:

$$\mathbb{E}\left[\sup_{f\in\mathcal{I}_\aleph}\sum_{i=1}^n \gamma_i\left(f(\boldsymbol{y}_i) - f(\boldsymbol{y}_i')\right)\right]$$

$$= \mathbb{E}\left[\sup_{||\boldsymbol{s}||\leq\aleph}\sum_{i=1}^n \gamma_i\left(\boldsymbol{s}^\top(\boldsymbol{y}_i - \boldsymbol{y}_i')\right)\right]$$

$$\leq \aleph\cdot\mathbb{E}\left[\left|\left|\sum_i \gamma_i(\boldsymbol{y}_i - \boldsymbol{y}_i')\right|\right|\right]$$

$$\leq \aleph\left(\mathbb{E}\left[\left|\left|\sum_i \gamma_i(\boldsymbol{y}_i - \boldsymbol{y}_i')\right|\right|^2\right]\right)^{1/2}$$

$$= \aleph\left(\sum_i \left|\left|(\boldsymbol{y}_i - \boldsymbol{y}_i')\right|\right|^2\right)^{1/2}$$

$$= \aleph\cdot\|\boldsymbol{y} - \boldsymbol{y}'\|.$$

This suggests that $Q(\mathcal{I}_\aleph) \leq \aleph$.

$\square$

**Proof of Theorem 3**: Thm.3 directly follows Lem.1-Lem.3.

### B.4 Proof of Theorem 4

Our proof requires the following three lemmas from [Golub and Van Loan, 2012].

**Lemma 4.** *Let $\boldsymbol{A} \in \mathbb{R}^{m\times n}$ with $m \leq n$. Moreover, denote the singular values of $\boldsymbol{A}$ as $\sigma_1(\boldsymbol{A}) \geq \sigma_2(\boldsymbol{A}) \geq \cdots \geq \sigma_m(\boldsymbol{A})$. Let $\boldsymbol{B} = \begin{bmatrix} \boldsymbol{0} & \boldsymbol{A} \\ \boldsymbol{A}^\top & \boldsymbol{0} \end{bmatrix}$. The eigenvalues of $\boldsymbol{B}$ are $\lambda_1(\boldsymbol{B}) \geq \lambda_2(\boldsymbol{B}) \geq \cdots \geq \lambda_{m+n}(\boldsymbol{B})$. We have : $\lambda_1(\boldsymbol{B}) = \sigma_1(\boldsymbol{A}), \cdots, \lambda_m(\boldsymbol{B}) = \sigma_m(\boldsymbol{A}), \lambda_{m+1}(\boldsymbol{B}) = \lambda_{m+2}(\boldsymbol{B}) = \cdots = \lambda_n(\boldsymbol{B}) = 0, \lambda_{n+1}(\boldsymbol{B}) = -\sigma_m(\boldsymbol{A}), \lambda_{n+2}(\boldsymbol{B}) = -\sigma_{m-1}(\boldsymbol{A}), \cdots, \lambda_{n+m}(\boldsymbol{B}) = -\sigma_1(\boldsymbol{A})$.*

**Lemma 5.** *Let $\boldsymbol{A}, \boldsymbol{B} \in \mathbb{S}^N$, $\lambda_1(\boldsymbol{A}) \geq \lambda_2(\boldsymbol{A}) \geq \cdots \geq \lambda_N(\boldsymbol{A})$; $\lambda_1(\boldsymbol{B}) \geq \lambda_2(\boldsymbol{B}) \geq \cdots \geq \lambda_N(\boldsymbol{B})$. Then:*

$$\lambda_N(\boldsymbol{A}) + \lambda_j(\boldsymbol{B}) \leq \lambda_j(\boldsymbol{A}+\boldsymbol{B}) \leq \lambda_1(\boldsymbol{A}) + \lambda_j(\boldsymbol{B}), \text{ for } j = 1, 2, \cdots, i.$$

**Lemma 6** (Wielandt-Hoffman). *For $\boldsymbol{A} \in \mathbb{S}^N$, and $\boldsymbol{E} \in \mathbb{S}^N$, we have:*

$$\sum_{i=1}^N (\lambda_i(\boldsymbol{A}+\boldsymbol{E}) - \lambda_i(\boldsymbol{A}))^2 \leq \|\boldsymbol{E}\|_F^2.$$

*Moreover, for $\widetilde{\boldsymbol{A}} \in \mathbb{R}^{m\times n}$, and $\widetilde{\boldsymbol{E}} \in \mathbb{R}^{m\times n}$, we have:*

$$\sum_{i=1}^N \left(\sigma_i\left(\widetilde{\boldsymbol{A}}+\widetilde{\boldsymbol{E}}\right) - \sigma_i\left(\widetilde{\boldsymbol{A}}\right)\right)^2 \leq \|\widetilde{\boldsymbol{E}}\|_F^2.$$

*Proof.*
**proof of (a)** According to the definition of $\Delta(\boldsymbol{S}^\ddagger)$ and $\Delta(\boldsymbol{S})$, we have :

$$\left|\left|T\Delta(\boldsymbol{S}^\ddagger) - \Delta(\boldsymbol{S})\right|\right|_F \leq \left|\left|\begin{bmatrix} \boldsymbol{0} & T\boldsymbol{S}^\ddagger \\ T\boldsymbol{S}^{\ddagger\top} & \boldsymbol{0} \end{bmatrix} - \begin{bmatrix} \boldsymbol{0} & \boldsymbol{S} \\ \boldsymbol{S}^\top & \boldsymbol{0} \end{bmatrix}\right|\right|_F + \left|\left|T\boldsymbol{S}^\ddagger\boldsymbol{1}_k - \boldsymbol{S}\boldsymbol{1}_k\right|\right|_2 + \left|\left|T\boldsymbol{S}^{\ddagger^\top}\boldsymbol{1}_T - \boldsymbol{S}^\top\boldsymbol{1}_T\right|\right|_2$$

$$= \sqrt{2}\left|\left|T\boldsymbol{S}^\ddagger - \boldsymbol{S}\right|\right|_F + \left|\left|T\boldsymbol{S}^\ddagger\boldsymbol{1}_k - \boldsymbol{S}\boldsymbol{1}_k\right|\right|_2 + \left|\left|T\boldsymbol{S}^{\ddagger^\top}\boldsymbol{1}_T - \boldsymbol{S}^\top\boldsymbol{1}_T\right|\right|_2$$

$$\leq \sqrt{2\xi_2} + \left|\left|\boldsymbol{S} - T\boldsymbol{S}^\ddagger\right|\right|_F \cdot (\|\boldsymbol{1}_k\|_2 + \|\boldsymbol{1}_T\|_2)$$

$$\leq \sqrt{2\xi_2} + \sqrt{\xi_2}(\sqrt{k} + \sqrt{T}).$$

According to Lem.6, we have:

$$\sum_{i=N-K+1}^{N} \lambda_i(\Delta(\boldsymbol{S})) \leq \sum_{i=N-K+1}^{N} T\lambda_i(\Delta(\boldsymbol{S}^{\ddagger})) + \sqrt{2\xi_2 K} + \sqrt{\xi_2 K}(\sqrt{k} + \sqrt{T})$$
$$= T\xi_3 + \sqrt{2\xi_2 K} + \sqrt{\xi_2 K}(\sqrt{k} + \sqrt{T}).$$

**proof of (b)** Let $\boldsymbol{A} = diag(\boldsymbol{S}^{\ddagger}\mathbf{1})$, $\boldsymbol{B} = -\Delta(\boldsymbol{S}^{\ddagger})$, we have $\boldsymbol{A} + \boldsymbol{B} = \boldsymbol{A}_{\boldsymbol{l} \cup \boldsymbol{o}}$. Applying Lem. 5 for $j = 1, 2, \cdots, K$ we have :

$$\lambda_N \left(diag(\boldsymbol{S}^{\ddagger}\mathbf{1})\right) - \lambda_{N-j+1}\left(\Delta(\boldsymbol{S}^{\ddagger})\right) \leq \lambda_j \left(\boldsymbol{A}_{\boldsymbol{l}\cup\boldsymbol{o}}\right) \overset{(\star)}{=} \sigma_j(\boldsymbol{S}^{\ddagger}) \leq \lambda_1 \left(diag(\boldsymbol{S}^{\ddagger}\mathbf{1})\right) - \lambda_{N-j+1}\left(\Delta(\boldsymbol{S}^{\ddagger})\right), \tag{7}$$

where $(\star)$ is due to Lem.4. According to Thm.2 and the definition of $\xi_3$, we have:

$$\sum_{i=1}^{K} \lambda_{N-i+1}\left(\Delta(\boldsymbol{S}^{\ddagger})\right) \leq \langle\Delta(\boldsymbol{S}^{\ddagger}), \boldsymbol{U}\rangle \leq \xi_3. \tag{8}$$

With Lem.4 and Eq.(7)-(8), we have :

$$\frac{1}{T} - \xi_3 \leq \sigma_i(\boldsymbol{S}^{\ddagger}) \leq \frac{1}{k}, \text{ for } i = 1, 2, \cdots K.$$

Since $d(\boldsymbol{S}, T\boldsymbol{S}^{\ddagger}) \leq \xi_2$, according to Lem.6, we have:

$$T\sigma_i\left(\boldsymbol{S}^{\ddagger}\right) - \sqrt{\xi_2} \leq \sigma_i\left(\boldsymbol{S}\right) \leq T\sigma_i\left(\boldsymbol{S}^{\ddagger}\right) + \sqrt{\xi_2}, \; i = 1, 2, \cdots K.$$

The theorem follows from Thm.4.2.3 in [Horn and Johnson, 2012] and the fact that $\sigma_i(\boldsymbol{S}) = \sqrt{\lambda_i(\boldsymbol{S}^{\top}\boldsymbol{S})}$.

$\square$

## B.5   Proof of Theorem 5

First, we need the following two lemmas about the sine $\Theta$ theorem, the proof of which could be found in [Yin and Shen, 2018, Lem.1] and [Yu et al., 2014, Thm.1] respectively.

**Lemma 7.** *Let $\boldsymbol{X}$ and $\boldsymbol{Y}$ be two orthogonal matrices of $\mathbb{R}^{n \times n}$. Let $\boldsymbol{X} = [\boldsymbol{X}_0, \boldsymbol{X}_1]$ and $\boldsymbol{Y} = [\boldsymbol{Y}_0, \boldsymbol{Y}_1]$, where $\boldsymbol{X}_0$ and $\boldsymbol{Y}_0$ are the first $K$ columns of $\boldsymbol{X}$ and $\boldsymbol{Y}$, respectively. Then, we have:*

$$\|\boldsymbol{X}_0\boldsymbol{X}_0^{\top} - \boldsymbol{Y}_0\boldsymbol{Y}_0^{\top}\|_F \leq \sqrt{2}\|\boldsymbol{X}_0^{\top}\boldsymbol{Y}_1\|_F.$$

**Lemma 8** (sine $\Theta$). *Let $\boldsymbol{\Sigma}, \hat{\boldsymbol{\Sigma}}$ be symmetric with eigenvalues $\lambda_1 \geq \cdots, \lambda_p$ and $\hat{\lambda_1} \cdots \hat{\lambda_p}$, respectively. Fix $1 \leq K \leq p$, and let $\boldsymbol{X}_0 = [\boldsymbol{v}_1, \boldsymbol{v}_2, \cdots, \boldsymbol{v}_K] \in \mathbb{R}^{p \times K}$ and $\hat{\boldsymbol{Y}_0} = [\hat{\boldsymbol{v}}_1, \hat{\boldsymbol{v}}_2, \cdots, \hat{\boldsymbol{v}}_K]$ and let $\boldsymbol{X}_1 = [\boldsymbol{v}_{K+1}, \cdots, \boldsymbol{v}_p]$ and $\boldsymbol{Y}_1 = [\hat{\boldsymbol{v}}_{K+1}, \cdots, \hat{\boldsymbol{v}}_p]$. For $1 \leq j \leq p$, we have $\boldsymbol{\Sigma}\boldsymbol{v}_j = \lambda_j\boldsymbol{v}_j$ and $\hat{\boldsymbol{\Sigma}}\hat{\boldsymbol{v}}_j = \hat{\lambda}_j\hat{\boldsymbol{v}}_j$. If $\delta = |\hat{\lambda}_{K+1} - \lambda_K| > 0$, we have:*

$$\|\boldsymbol{X}_0^{\top}\boldsymbol{Y}_1\|_F \leq \frac{\|\boldsymbol{\Sigma} - \hat{\boldsymbol{\Sigma}}\|_F}{\delta}.$$

Now we are ready to prove the theorem.

*Proof.* For all $\boldsymbol{S}^{\star} \in \Pi(\mathbf{a}, \mathbf{b})$, such that $Supp(\boldsymbol{S}^{\star}) = \mathcal{G}$, according to Thm.2, we have that $V_K^{\star} = [\boldsymbol{f}_1^{\star}, \cdots, \boldsymbol{f}_{k+T}^{\star}]^{\top}$ the eigenvectors associated with the bottom $K$ eigenvalues of the Laplacian $\Delta(\boldsymbol{S}^{\star})$ much satisfy that $j \leq k$, $[\boldsymbol{f}_j^{\star}]_g = 1$ only if $\boldsymbol{l}_j$ belongs to group $g$ and that for $j > k$, $[\boldsymbol{f}_j^{\star}]_g = 1$ only if $\boldsymbol{o}_j$ belongs to group $g$. Moreover, we define $V_K = [\boldsymbol{f}_1, \cdots, \boldsymbol{f}_{k+T}]^{\top}$ as the eigenvectors associated with the bottom $K$ eigenvalues of $\Delta(\boldsymbol{S}^{\ddagger})$ and define

$$\mathcal{D}_{i,j}^{\star} = \|\boldsymbol{f}_i^{\star} - \boldsymbol{f}_{k+j}^{\star}\|_2^2, \; \mathcal{D}_{i,j} = \|\boldsymbol{f}_i - \boldsymbol{f}_{k+j}\|_2^2.$$

Since $S^\ddagger$ is selected from $\mathcal{H}$, according to Thm.2, we have:

$$\langle \mathcal{D}, S^\ddagger \rangle = \sum_{i=N-K+1}^{N} \lambda_i \left( \Delta(S^\ddagger) \right) \leq \langle U, \Delta(S^\ddagger) \rangle, \ \ \forall U \in \mathcal{M}.$$

This implies $\langle \mathcal{D}, S^\ddagger \rangle \leq \xi_3$.

Since for all $i$, $f_i^*$ is a 0-1 group indicator, we have

$$\mathcal{D}_{ij}^* = \begin{cases} 0, & l_i \text{ and } o_j \text{belong to the same group} \\ 2, & \text{otherwise} \end{cases}$$

and thus $\langle \mathcal{D}^*, S^\ddagger \rangle = 2\|S^{\ddagger supp^c}\|_1$. Then, we have :

$$\|S^{\ddagger supp^c}\|_1 \leq \frac{1}{2} \cdot \left( \xi_3 + \langle \mathcal{D}^\star - \mathcal{D}, S^\ddagger \rangle \right).$$

It now only remains to give an upper bound for $\langle \mathcal{D}^* - \mathcal{D}, S^\ddagger \rangle$. In fact:

$$\langle \mathcal{D}^\star - \mathcal{D}, S^\ddagger \rangle = \langle U^\star - U, \Delta(S^\ddagger) \rangle \leq \|\Delta(S^\ddagger)\|_F \cdot \|U - U^\star\|_F \leq \frac{\sqrt{\frac{2}{k} + \frac{6}{T}} \cdot \epsilon}{\lambda_{K+1}(\Delta(S^\ddagger))},$$

where $U^\star = V_K^\star V_K^{\star\top}, U = V_K V_K^\top$, and the last inequality follows from that

$$\|\Delta(S^\ddagger)\|_F = \sqrt{\left\| diag(\frac{\mathbf{1}_k}{k} + \frac{\mathbf{1}_T}{T}) - \begin{bmatrix} \mathbf{0} & S^\ddagger \\ S^{\ddagger\top} & \mathbf{0} \end{bmatrix} \right\|_F^2}$$

$$= \sqrt{\left\| \frac{\mathbf{1}_k}{k} \right\|_F^2 + \left\| \frac{\mathbf{1}_T}{T} \right\|_F^2 + 2\left\| S^\ddagger \right\|_F^2}$$

$$\leq \sqrt{\frac{1}{k} + \frac{1}{T} + \frac{2}{T^2}}.$$

and Lem.7 and Lem.8, with $\hat{\Sigma} = \Delta(S^\ddagger), \Sigma = \Delta_{S^\star}$, and $|\hat{\lambda}_{K+1} - \lambda_K| = \lambda_{K+1}(\Delta(S^\ddagger))$. This leads to an upper bound of $\|S^{\ddagger supp^c}\|_1$:

$$\|S^{\ddagger supp^c}\|_1 \leq \frac{1}{2} \cdot \left( \xi_3 + \frac{\sqrt{\frac{2}{k} + \frac{6}{T}} \cdot \epsilon}{\lambda_{K+1}(\Delta(S^\ddagger))} \right).$$

The final upper bound for $\|S^{\ddagger supp^c}\|_1$ follows that:

$$\epsilon \leq \sup_{S,S' \in \Pi(\mathbf{a},\mathbf{b})} \|\Delta(S) - \Delta(S')\|_F \leq 2\sqrt{2} \cdot \sup_{S \in \Pi(\mathbf{a},\mathbf{b})} \|S\|_F \leq 2\sqrt{\frac{2}{T}}.$$

$\square$

## C    Details of the Optimization Algorithm

**Solving the $L$ subproblem.**

Recall the $L$ subproblem of ($Obj$), where we fix all the other variable and solve $L$. This subproblem could be formulated as:

$$\underset{L}{\mathrm{argmin}} \ \sum_{i=1}^{T} \sum_{j=1}^{n_j} \mathcal{J}(Y^{(i)}, X^{(i)} L S^{(i)}) + \frac{\alpha_1}{2} \cdot ||L||_F^2.$$

Recall our setting in the experiments, we set $\mathcal{J}(Y^{(i)}, X^{(i)} L S^{(i)}) = \frac{1}{N_i}\|Y^{(i)} - X^{(i)} L S^{(i)}\|_F^2$ for regression problems, where $N_i$ is the number of instances for the $i$-th task. And we adopt the squared surrogate loss for AUC:

$$\mathcal{J}(Y^{(i)}, X^{(i)} L S^{(i)})_{LS} = \sum_{x_p \in \mathcal{S}_{+,i}} \sum_{x_q \in \mathcal{S}_{-,i}} \frac{s\left( g^{(i)}(x_p) - g^{(i)}(x_q) \right)}{n_{+,i} n_{-,i}}$$

for classification problem, where $g^{(i)}(\boldsymbol{x}) = \langle \boldsymbol{LS}^{(i)}, \boldsymbol{x} \rangle$, $\mathcal{S}_{+,i}$ is the set of positive instances for the $i$-th task, $\mathcal{S}_{-,i}$ is the set of negative instances for the $i$-th task. To simplify the squared surrogate AUC loss, we build an AUC graph, where the vertexes are the instances and the edges are only activated across different classes. Specifically, for each task $i$, we define the graph as $\mathcal{G}^{(i)}_{AUC} = (\mathcal{V}^{(i)}, \mathcal{E}^{(i)}, \mathcal{W}^{(i)})$. The vertex set $\mathcal{V}^{(i)}$ is the set of all the instances in $(\boldsymbol{X}^{(i)}, \boldsymbol{y}^{(i)})$. There exists an edge $(k,m) \in \mathcal{E}^{(i)}$ with weight $\mathcal{W}^{(i)}_{km} = \frac{1}{n_{+,i}n_{-,i}}$ if and only if $y^{(i)}_k \neq y^{(i)}_m$. Given $\mathcal{W}^{(i)}$, the Laplacian matrix $\Delta^{(i)}_{AUC}$ of $\mathcal{G}^{(i)}_{AUC}$ could be expressed as: $\Delta^{(i)}_{AUC} = diag(\mathcal{W}^{(i)}\mathbf{1}) - \mathcal{W}^{(i)}$. With the definition of $\Delta^{(i)}_{AUC}$, we could reformulate the empirical loss $\mathcal{J}(\boldsymbol{Y}^{(i)}, \boldsymbol{X}^{(i)}\boldsymbol{LS}^{(i)})_{AUC}$ as :

$$\mathcal{J}(\boldsymbol{Y}^{(i)}, \boldsymbol{X}^{(i)}\boldsymbol{LS}^{(i)})_{AUC} = (\boldsymbol{Y}^{(i)} - \boldsymbol{X}^{(i)}\boldsymbol{LS}^{(i)})^\top \Delta^{(i)}_{AUC}(\boldsymbol{Y}^{(i)} - \boldsymbol{X}^{(i)}\boldsymbol{LS}^{(i)}).$$

The gradient of $\boldsymbol{L}$ with these two loss functions are then given as:

$$\nabla_{\boldsymbol{L}}(\mathcal{J}_{LS}) = \sum_{i=1}^{T} \frac{1}{N_i} \left( \boldsymbol{X}^{(i)\top}\boldsymbol{X}^{(i)}\boldsymbol{LS}^{(i)}\boldsymbol{S}^{(i)\top} - \boldsymbol{X}^{(i)\top}\boldsymbol{Y}^{(i)}\boldsymbol{S}^{(i)\top} \right) + \alpha_1 \boldsymbol{L},$$

$$\nabla_{\boldsymbol{L}}(\mathcal{J}_{AUC}) = \sum_{i=1}^{T} \left( \boldsymbol{X}^{(i)\top}\Delta^{(i)}_{AUC}\boldsymbol{X}^{(i)}\boldsymbol{LS}^{(i)}\boldsymbol{S}^{(i)\top} - \boldsymbol{X}^{(i)\top}\Delta^{(i)}_{AUC}\boldsymbol{Y}^{(i)}\boldsymbol{S}^{(i)\top} \right) + \alpha_1 \boldsymbol{L}.$$

Though both losses enjoy a closed-form solution, it comes with an extremely high time complexity of $O(k^6 T^6)$. We adopt L-BFGS [Zhu et al., 1997] as our optimizer, which only requires loss and gradient evaluations. If $\boldsymbol{X}^{(i)\top}\boldsymbol{X}^{(i)}$, $\boldsymbol{X}^{(i)\top}\boldsymbol{Y}^{(i)}$, $\boldsymbol{X}^{(i)\top}\Delta^{(i)}_{AUC}\boldsymbol{X}^{(i)}$ and $\boldsymbol{X}^{(i)\top}\Delta^{(i)}_{AUC}\boldsymbol{Y}^{(i)}$ are precomputed and cached to the memory, we come to a complexity of $O(d^2 T + kdT)$ per gradient evaluation.

**Solving the $S$ subproblem** With the other parameters fixed, $\boldsymbol{S}$ could be solved from the following problem:

$$\underset{\boldsymbol{S}}{\operatorname{argmin}} \sum_{i=1}^{T} \sum_{j=1}^{n_j} \mathcal{J}(\boldsymbol{Y}^{(i)}, \boldsymbol{X}^{(i)}\boldsymbol{LS}^{(i)}) + \frac{\alpha_2}{2} \|\boldsymbol{S} - T\boldsymbol{S}^{\ddagger}\|_F^2.$$

With the squared loss for regression, the closed-form solution reads

$$\boldsymbol{S}^{(i)\star} = \left( \frac{\boldsymbol{A}_i^\top \boldsymbol{A}_i}{N_i} + \alpha_2 \boldsymbol{I} \right)^{-1} \left( \frac{\boldsymbol{A}_i^\top \boldsymbol{Y}^{(i)}}{N_i} + \alpha_2 T \boldsymbol{S}^{\ddagger} \right),$$

where $\boldsymbol{A}_i = \boldsymbol{X}^{(i)}\boldsymbol{L}$. Similarly, for the squared surrogate loss for AUC, the closed-form solution becomes:

$$\boldsymbol{S}^{(i)\star} = \left( \frac{\boldsymbol{A}_i^\top \Delta^{(i)}_{AUC} \boldsymbol{A}_i}{N_i} + \alpha_2 \boldsymbol{I} \right)^{-1} \left( \frac{\boldsymbol{A}_i^\top \Delta^{(i)}_{AUC} \boldsymbol{Y}^{(i)}}{N_i} + \alpha_2 T \boldsymbol{S}^{\ddagger} \right).$$

## D    Experiments

### D.1    Real-world Dataset

**Competitors** Now we briefly introduce competitors adopted in this paper. To show the improvement toward different types of methods, our method is compared with the following methods:

- RAMUSA [Han and Zhang, 2016] adopts a capped trace norm regularizer to minimize only the singular values smaller than an adaptively tuned threshold.

- GOMTL [Kumar and III, 2012] decomposes per-task parameters as linear combinations of latent task basis, where $\Omega(\boldsymbol{S})$ is set to $\ell_1$ penalty. In this way, it learns an arbitrary sparse LATM.

- CoCMTL [Xu et al., 2015] realizes the task-specific co-clustering via minimizing the truncated sum-of-squares of the singular values of the task matrix.

- CMTL [Jacob et al., 2009] assumes that the per-tasks parameters are clustered into a given number of groups. Specifically, it leverages a clustered model parameter via simultaneously encouraging a large between-cluster variance and a small within-cluster variance.

- NC-CMTL [Nie et al., 2018] explores shared information among different tasks with a non-convex low-rank spectral regularizer and a robust re-weighting scheme.

Figure 3: (a) Performance Comparison Curve over the Simulation Dataset, with varying training data ratio. (b) Performance Comparison over the Sun Dataset.

- `VSTGMTL` [Jeong and Jun, 2018] implements simultaneous variable selection and learning with a low-rank decomposition.

- `AMTL` [Lee et al., 2016] assumes that each task parameter is a linear combination of other tasks and leverages asymmetric transfer between tasks with a sparse selection on the asymmetric transfer matrix.

- `GAMTL` [Liu and Pan, 2017] Similar to `AMTL`, `GAMTL` also assumes that each task parameter is a linear combination of other tasks. Moreover, it also adopts the trace lasso penalty to maintain a better grouping structure.

**AWA2-Attribute**: Based on the `AWA-2` dataset [Xian et al., 2018], we construct an MTL dataset containing 85 tasks. Each task in this dataset is a binary classification to recognize whether a given attribute is presented in a given instance. To construct this dataset, for each attribute we sampled 50 positive instances and 150 negative instances. This yields to a total volume of 17,000 samples.
 **AWA2-Class**: Similar to AWA2-Attribute, we perform another MTL dataset concerning the animal class recognition. The $i$-th task in this dataset is a binary classification to recognize whether a given sample belongs to the $i$-th class in AWA2. Similarly, we sampled 50 positive instances and 150 negative instances for each task. This yields to a total volume of 10,000 samples.
**School Dataset** : This dataset is collected from the Inner London Education Authority, which consists of examination scores of 15362 students from 139 schools in London [Kumar and III, 2012]. Here, the score prediction for each school corresponds to a task, thus giving a total of 139 tasks.
**Shoes Dataset.** The Shoes Dataset [Kovashka and Grauman, 2015] is a popular attribute prediction benchmark, which consists of 14,658 online shopping shoe images with 7 attributes (BR: brown, CM: comfortable, FA: fashionable, FM: formal, OP: open, ON: ornate, PT: pointy). In this dataset, annotators with various knowledge are invited to judge whether a specific attribute is present in an image. Here, the tasks are then consisted of predicting the attribute annotations for different users. Specifically, each user is randomly assigned with 50 images, and there are at least 190 users for each attribute who take part in the process, which results in a total volume of 90,000 annotations.
**Sun Dataset.** The Sun Dataset [Kovashka and Grauman, 2015] contains 14,340 scene images from SUN Attribute Database [andJames Hays, 2012], with personalized annotations over 5 attributes (CL: Cluttered, MO: Modern, OP: Opening Area, RU: Rustic, SO: Soothe). With a similar annotating procedure, 64,900 annotations are obtained in this dataset. The tasks are defined in the same way as Shoes dataset.
**Pre-processing.** For AWA2-Attribute and AWA2-Attribute, we adopt the ILSVRC-pretrained ResNet101 feature which is used in [Xian et al., 2018]. For School Dataset, we use the features provided in the MALSAR package [Zhou et al., 2011b]. For Shoes dataset, we simply adopt the GIST and color histogram provided in [Kovashka et al., 2012] as input features, whereas we deploy the 2048-dim feature vectors extracted by the Inception-V3 [Szegedy et al., 2016] network for Sun's data. The reason leading us to two different feature extraction strategies lies in that the images in Shoes dataset are photographed on a white background, while images in Sun dataset usually suffer from much more complicated backgrounds. For the classification datasets, we perform PCA to reduce the redundancy of these features before training. For sun and shoes dataset, we notice that users who extremely prefer to provide merely one class of labels may lead to large biases. To e-

Table 1: Performance Comparison over General MTL Datasets (mean ± std)

| Algorithms | AWA2-Attr($\uparrow$) | AWA2-Cls ($\uparrow$) | School ($\downarrow$) |
|---|---|---|---|
| RAMUSA [Han and Zhang, 2016] | 88.60±0.66 | 93.06±0.53 | 10.52±0.09 |
| GOMTL [Kumar and III, 2012] | 89.56±0.33 | 88.22±1.18 | 10.26±0.11 |
| CoCMTL [Xu et al., 2015] | 92.29±0.35 | 94.69±0.73 | 12.06±0.09 |
| CMTL [Zhong and Kwok, 2012] | 92.95±0.35 | 94.81±0.70 | 12.06±0.09 |
| VSTGMTL [Jeong and Jun, 2018] | 89.31±0.39 | 92.03±0.94 | 10.17±0.08 |
| GAMTL [Liu and Pan, 2017] | 89.39±0.42 | 92.55±0.53 | 10.50±0.12 |
| NC-CMTL [Nie et al., 2018] | 92.99±0.32 | 95.10±0.60 | 10.53±0.12 |
| AMTL [Lee et al., 2016] | 92.15±0.34 | 95.76±0.44 | 12.15±0.09 |
| GBDSP | 92.73±0.29 | 97.86±0.22 | 10.10±0.08 |

liminate such effect, we manually remove users who give less than 8 annotations for the minority class.

**Performance comparison** The performance results over general MTL datasets are shown in Tab.1, and the results for personalized attribute prediction datasets are shown in Fig. 3(a) and Fig. 3(b). Then we could make the following observations: 1) Our proposed algorithm consistently outperforms all the competitors in all datasets except AWA2-Attr. Moreover, on AWA2-Attr, GBDSP shows competitive performance with CMTL and NC-CMTL. 2) Comparing with GOMTL, GBDSP always shows significant performance improvements. This implies that leveraging a block-diagonal rather than arbitrarily sparse LATM helps to improve the performance. 3) In most cases, we find AMTL outperforms the other low-rank constrained methods on all the datasets, as it explicitly models and reduces the influence of negative transfer via asymmetric learning. 4) Our proposed method outperforms AMTL on most results. One possible reason is that AMTL avoids negative transfer via differentiates the hard tasks with the easy tasks, while GBDSP provides a finer-grained against negative transfer via suppressing inter-group transfer with the latent task representation.

**Task Correlation study on AWA-Attr and AWA-Cls** To see how learning a block-diagonal $S$ helps to recover a valuable task correlation matrix, we compare the predicted correlation matrix with the semantic correlation matrix obtained from the class-attribute relations. Here the predicted task correlation matrix is calculated from $\widetilde{R} = |S|^\top |S|$, with $\widetilde{R}_{ij}$ measures the similarity between the latent task assignment between $o_i$ and $o_j$. The semantic correlation is obtained based on the predicate-matrix $P$ provided by the AW2 dataset [3], where $P \in \{0,1\}^{50 \times 85}$ and $P_{ij} = 1$ only if attribute $j$ is relevant to class $i$. For AWA-Attr, we formulate the semantic task correlation matrix $\bar{R}$ as $\bar{R}_{ij} = \sum_{k=1}^{50} \delta(P_{ki} = P_{kj})$. Here $\bar{R}_{ij}$ counts how many times attribute $i$ and $j$ are simultaneously relevant/irrelevant to the same class. For AWA-Cls, we have a similar construction of $\bar{R}$ as $\bar{R}_{ij} = \sum_{k=1}^{85} \delta(P_{ik} = P_{jk})$, where $\bar{R}_{ij}$ counts how many times class $i$ and $j$ are simultaneously relevant/irrelevant to the same attribute. Moreover, the matrices are normalized such that $\max_j \left\{ \widetilde{R}_{ij} \right\} = 1$ and $\max_j \left\{ \bar{R}_{ij} \right\} = 1$, $\forall i$.

In Fig.5 and Fig.6 , we visualize $\widetilde{R}$ generated by GOMTL and GBDSP respectively together with $\bar{R}$. Moreover, as shown in Fig.4, we also calculate the cosine similarity between the $\widetilde{R}$ and $\bar{R}$. The results indicate that learning a block-diagonal instead of an arbitrary sparse $S$ matrix helps to preserve the semantic similarity among tasks.

Figure 4: Task Correlation Comparison

**Fine-grained Comparisons on Shoes and Sun Dataset** Fig.7 shows attribute-wise comparison on Shoes and Sun Dataset.

Figure 5: Comparison over the correlation structure for AWA2-Attribute: a) $\widetilde{\boldsymbol{R}}$ obtained from GOMTL, (b) $\widetilde{\boldsymbol{R}}$ obtained from GBDSP (c) $\bar{\boldsymbol{R}}$

Figure 6: Comparison over the correlation structure for AWA2-Class: (a) $\widetilde{\boldsymbol{R}}$ obtained from GOMTL, (b) $\widetilde{\boldsymbol{R}}$ obtained from GBDSP (c) $\bar{\boldsymbol{R}}$

(a) Shoes Brown  (b) Shoes Comfortable  (c) Shoes Fashionable

(d) Shoes Formal  (e) Shoes Open  (f) Shoes Ornate

(g) Shoes Pointy  (h) Sun Cluttered  (i) Sun Modern

(j) Sun Open Area  (k) Sun Rustic  (l) Sun Smoothe

Figure 7: Attribute-wise Performance Comparisons on Shoes and Sun