[Reviews · NeurIPS 2019]

Reviewer 1



In this work, authors proposed a framework for multi-task learning, where there is assumed to be latent task space, and the learner simultaneously learn the latent task representation as well as the coefficients for these latent variables, namely the Latent Task Assignment Matrix (LTAM). Authors further imposed block-diagonal structure on the assignment matrix, and developed spectral regularizers for it. Authors then proposed a relaxed objective that can be optimized via a scheme similar to block coordinate descent. Authors also provided generalization learning guarantees as well as the structure recovery performance. Simulation experiments showed that the proposed algorithm can recover the true structure and provide improvement in prediction accuracy. The framework is interesting and the results are meaningful. However, the paper is poorly written and hard to follow. The authors didn't give enough explanation and discussion of their results, which makes them hard to digest. I think the paper needs major revision. Some questions and comments: - This whole framework is limited to linear models. Can this be easily extended to other hypotheses families? - About optimization: the overall objective is non-increasing, which is nice, can we say anything about convergence rate? Also, since the objective is not jointly convex, what happens if the algorithm converges to a local minimum instead of a global one? - In the definition of H(L, S, \tilde S, U), is there any restriction on U and \tilde S? I thought they must belong to some sets, as in the definition of Obj, but these are not mentioned in the definition of H. - I can't find the proof of Theorem 3, which seems to be adapted from another paper. However I'd still like to see how the adaption is made, just to double check. Also, there is no \xi_3 in the generalization bound, which seems strange since it also defines the hypothesis set H. ------ I read other reviews and authors' response, and most of my concerns are addressed. I'll increase the score to 6. However, I still think the paper can benefit a lot from adding more explanations for the motivation and for the theoretical results, e.g. those given in the response.

Reviewer 2



1. This paper is clearly written. The methodology section covers dense information. However, the subtitles before each paragraph make it easy to read. 2. This paper formulates a regularization scheme for the negative transfer problem. The arguments in this paper follow rigorous math,which makes the main idea convincing. 3. The figures in this paper are nicely plotted. Fig.1 demonstrates the structure recovery properties of their proposed regularization term. Fig.2 shows that the spectral embedding does has a strong semantic power as promised in the paper. 4. A minor issue is that some details of the optimization process are missing. More elaborations should be provided to improve clarity. (see also the Improvements part)

Reviewer 3



1) This paper is well-written, the goal and motivation are well-clarified and convincing. Overall it is easy to follow. 2) It is very nice to see that the methodology is strictly organized under solid theoretical analysis and discussion. Moreover, we see sufficient empirical studies as well, where we can observe some interesting results and visualization. 3) Last but not least, I have to say that what attracts me most is the insights this paper brings to me. It does not merely give a new regularizer for leveraging heterogenous block diagonal structure. More importantly, it also brings a new wind by bridging the Laplacian-based structural learning problem with the optimal transport problem. In my opinion, this suggests that leveraging the expected structural sparsity could then be regarded as a matching problem across latent tasks and output tasks.

[Author Response · NeurIPS 2019]

# Response for "Generalized Block-Diagonal Structure Pursuit: Learning Soft Latent Task Assignment against Negative Transfer"
## ID 3136

We thank all the reviewers for their valuable comments. We have fixed the typos pointed out by the reviewers. Below are the responses to each reviewer.

**To Reviewer 1**

**Q1**: **Is the framework limited only to linear models?** Our framework could be applied to more general model families in view of the following two perspectives. By the formulation $\boldsymbol{W} = \boldsymbol{LS}$, our framework provides a latent representation of the task weights $\boldsymbol{W}$. It could be naturally extended to more complicated models such as DNNs via modifying the latent expression $\boldsymbol{L}$. From the generative point-of-view, our framework could also be extended to cover more complicated distribution families via changing $g_1(\cdot)$, $g_2(\cdot)$ and $g_3(\cdot)$.

**Q2**: **The convergence property and the convergence rate of the algorithm? What if only a local solution is found? 1)** Under mild assumptions, we can say that the proposed algorithm enjoys the global convergence property defined for non-convex problems, where *both the objective sequence and the parameter sequence converge to a critical point*. Moreover the convergence rate should be $\mathsf{O}(\frac{1}{T})$, which is *sublinear*. **2)** For non-convex problems, global convergence to a critical point is the best we can do in a general sense. The global minimum could be found only when the initial point is located in a local convex landscape covering the optimal solution. Nonetheless, according to Thm.3, the generalization ability will be promising if the loss is small (not necessarily only the optimal value) and the hypothesis space is properly chosen. In this sense, a local critical point would be a good candidate solution. This has also been suggested by our experimental results.

**Q3**: **Are the constraints in the Obj included in the class $\mathcal{H}(\boldsymbol{L}, \boldsymbol{S}, \widetilde{\boldsymbol{S}}, \boldsymbol{U})$?** The constraints are included in the hypothesis space. We will include them in the new version.

**Q4**: **The proof of Thm.3?** The proof follows naturally from Lem.3 and Thm. 13 in Ref.[23] mentioned in the main paper. There are only two key differences. We assume that $l(y, \cdot) : \hat{y} \mapsto [0, M]$ instead of $[0, 1]$. That is why we use $\frac{\Delta}{M}$ in our case. Moreover, we assume that $\breve{\ell}(\cdot) = l(y, \cdot)/M$ is $\phi$-Lipschitz continuous, instead of 1-Lipschitz continuous. That is why the first two terms (which gives an upper bound for the Gaussian Average of $\breve{\ell} \circ \left(\mathcal{H}(\boldsymbol{L}, \boldsymbol{S}, \widetilde{\boldsymbol{S}}, \boldsymbol{U})\right)$) on the right hand side are multiplied by $\phi$. We will provide a detailed proof in the new version.

**Q5**: $\xi_3$ **does not appear in Thm.3, how does it benefit the hypothesis space?** $\xi_3$ benefits the hypothesis space in the following sense. According to Thm.4, we see that decreasing $\xi_3$ reduces the sensitivity of numerical perturbation on the principle components of $\boldsymbol{S}$. More importantly, Thm.5 shows that shrinking $\xi_3$ results in a better recovery of the desired structure and helps to overcome the negative transfer issue. From the generalization perspective, the expected structure is sparse in most cases, this leads to an implicit control of the $\mathcal{VC}$ dimension of the space, which suggests that $\xi_3$ also contributes to the generalization ability.

**To Reviewer 2**

**Q1**: **For $\widetilde{S}$ subroutine, why adopt the dual problem? More discussion on the barycenter projection mapping.** 1) The dual problem could be solved much more efficiently than the primal. The dual problem only contains $\mathsf{O}(k + T)$ parameters while the primal requires $\mathsf{O}(kT)$ parameters. 2) Since $\widetilde{\boldsymbol{S}}$ is the solution of the regularized OT problem, we have $\widetilde{\boldsymbol{S}}_{ij} = \mathbb{P}(\boldsymbol{l} = i, \boldsymbol{o} = j)$. Then $\mathbb{E}_{\boldsymbol{l}|\boldsymbol{o}=i}(\boldsymbol{L}) = \frac{\boldsymbol{L}\widetilde{\boldsymbol{S}}^{(i)}}{T}$. In this sense, $\frac{\boldsymbol{L}\widetilde{\boldsymbol{S}}^{(i)}}{T}$ represents the output task as a barycenter in the latent task embedding space, since $\frac{\boldsymbol{L}\widetilde{\boldsymbol{S}}^{(i)}}{T} = \mathsf{argmin}_{\boldsymbol{z}} \, \mathbb{E}_{\boldsymbol{l}|\boldsymbol{o}=i}(d(\boldsymbol{L}^{(i)}, \boldsymbol{z}))$.

**Q2**: **The phrasing in Thm.2.** We will correct the phrasing issue as you suggested.

**To Reviewer 3**

**Q1**: **Discussion on Theorem 3.** Thm.3 states that if the magnitude of $\boldsymbol{COV}(\boldsymbol{X})$ is small, the difference between the population version of the task-averaged risk and the empirical risk $\Delta$ tends to zero asymptotically when $n \to +\infty$. Here $\boldsymbol{COV}(\boldsymbol{X})$ captures the correlation of the data points of all the samples. Please see the Reference [23] in the main paper for more details.

**Q2**: **Error bar in Fig.2.** The error bar in Fig.2 represents the standard deviation over all repetitions.

**Q3**: **Real-world dataset should be mentioned at the main paper.** We will add a brief introduction of the real-world dataset in a new version.

[Meta-Review · NeurIPS 2019]

The authors present a novel framework to address the issue of negative transfer when conducting multi-task learning. All reviewers agree that the setting is useful and that the algorithm is well motivated and has demonstrated value. The main concern is around clarity of some parts of the paper and the ease of interpreting some of the theoretical results. The author feedback has addressed most of the questions. I strongly expect the final version is expected to include these clarifications as well as much additional polishing as possible.